# Regret-Free Reinforcement Learning for Temporal Logic Specifications

**Rupak Majumdar** [1]  **Mahmoud Salamati** [1]  **Sadegh Soudjani** [1 2]

## Abstract

Learning to control an unknown dynamical system with respect to high-level temporal specifications is an important problem in control theory. We present the first regret-free online algorithm for learning a controller for linear temporal logic (LTL) specifications for systems with unknown dynamics. We assume that the underlying (unknown) dynamics is modeled by a finite-state and action Markov decision process (MDP). Our core technical result is a regret-free learning algorithm for infinite-horizon reach-avoid problems on MDPs. For general LTL specifications, we show that the synthesis problem can be reduced to a reach-avoid problem once the graph structure is known. Additionally, we provide an algorithm for learning the graph structure, assuming knowledge of a minimum transition probability, which operates independently of the main regret-free algorithm. Our LTL controller synthesis algorithm provides sharp bounds on how close we are to achieving optimal behavior after a finite number of learning episodes. In contrast, previous algorithms for LTL synthesis only provide asymptotic guarantees, which give no insight into the transient performance during the learning phase.

## 1. Introduction

We consider the problem of learning an optimal control policy for a stochastic system, whose dynamics are unknown, with respect to linear temporal logic (LTL) specifications. This is a core problem in control and robotics, with a rich body of existing techniques. A fundamental approach to the problem is to apply reinforcement learning (RL): the agent maintains a model of the world learned through exploration, and computes a sequence of approximations to an optimal control policy that maximizes the probability that the temporal specification is satisfied (Icarte et al., 2018; Camacho et al., 2019; Hasanbeig et al., 2019; Kazemi et al., 2022; Hahn et al., 2019; Oura et al., 2020; Cai et al., 2021; Bozkurt et al., 2021; Sickert et al., 2016; Alur et al., 2022; Fu & Topcu, 2014; Voloshin et al., 2022). In the limit, the approximations converge to the optimal policy.

In practice, it is not enough to know that approximations converge in the limit; we would prefer to know how close our current policy is to the optimal one, and to stop learning when the policy is close to optimal. One way to quantify this is through *regret minimization* (Auer et al., 2008; Agarwal et al., 2014; Srinivas et al., 2012; Dann et al., 2017). For an online learning algorithm, intuitively, regret is defined as the difference between the accumulated rewards collected by an optimal policy and the algorithm during learning. A learning algorithm is called *regret-free* if the regret grows sublinearly with the number of episodes.

In this paper, we propose the first regret-free learning algorithm for policy synthesis against LTL objectives. We consider the class of systems whose dynamics can be captured by a finite-state and action Markov decision process (MDP) with unknown transition probabilities and *fixed* initial state $s_{\text{init}}$. The objective is to synthesize a policy that maximizes the probability of satisfying a given LTL specification $\varphi$. Let $\pi^*$ denote an optimal policy, meaning that applying $\pi^*$ maximizes the satisfaction probability for $\varphi$. Our online algorithm produces a sequence of policies $\pi_1, \pi_2, \ldots$, one per episode. Let $v^*(s_{\text{init}})$ and $v_k(s_{\text{init}})$ denote, respectively, the satisfaction probabilities of $\pi^*$ and $\pi_k$ when starting from $s_{\text{init}}$. After passing $K > 0$ episodes, we define the regret as $R(K) := \sum_{k=1}^{K}(v^*(s_{\text{init}}) - v_k(s_{\text{init}}))$, the accumulated difference between the satisfaction probabilities under the optimal and learned policy. Our algorithm ensures that $\lim_{K \to \infty} R(K)/K = 0$.

We first state our algorithm for *reach-avoid* specifications, the subset of LTL specifications that require a set of goal states must be visited while avoiding hitting a set of bad states. In every episode, our algorithm (1) computes an *interval MDP* (iMDP) by taking all the collected state observations into account; (2) finds an *optimistic* policy over the computed iMDP to solve the reach-avoid problem; and (3) executes the computed policy before an *episode-specific deadline* is reached. We prove that the regret of our al-

[1]MPI-SWS, Kaiserslautern, Germany [2]University of Birmingham, Birmingham, UK. Correspondence to: Mahmoud Salamati <msalamati@mpi-sws.org>.

*Proceedings of the $42^{nd}$ International Conference on Machine Learning*, Vancouver, Canada. PMLR 267, 2025. Copyright 2025 by the author(s).

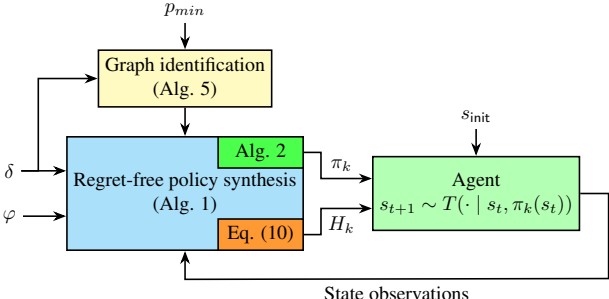

Figure 1: Overview of the approach: the LTL specification $\varphi$ along with the graph learned through the application of Alg. 5 are used to reduce the general synthesis problem into a reach-avoid problem. The confidence parameter $\delta \in (0,1)$ and the lower bound on the minimum transition probability $p_{min} \in (0,1)$ are used to compute the regret bound for Alg. 1 and the sample complexity for Alg. 5, respectively. Applying Alg. 1 yields a policy $\pi_k$ (using Alg. 2) and an episode length $H_k$ (using Eq. (7)), both of which are passed to the agent for execution in episode $k \geq 1$. The agent, modeled as a finite MDP with transition function $T$, executes the policy $\pi_k$ for at most $H_k$ time steps starting at state $s_{\text{init}}$, and passes the observed transitions back to the policy.

gorithm grows sublinearly with respect to the number of episodes.

Next, for a general LTL specification, we show that the synthesis problem can be reduced to a reach-avoid problem if we know the graph structure of the MDP. Finally, we provide a polynomial algorithm for learning the graph structure of the MDP based only on having a positive lower bound on its non-zero transition probabilities. Fig. 1 provides an overview of our proposed online policy learning algorithm. Policy learning can be stopped after $k^* \geq 1$ episodes if the average regret $R(k^*)/k^*$ becomes sufficiently small, indicating that the learned policy's satisfaction probability is nearly optimal on average, with confidence at least $1 - \delta$. This can be verified using our proposed regret bounds.

There exist regret-free algorithms for solving infinite-horizon policy synthesis problems, but they cannot be applied to our problem as we will discuss in more detail in Sec. 2. Our primary objective is to minimize the regret with respect to an optimal policy that maximizes the *satisfaction probability* of a specification. The regret for each episode is naturally bounded by one, which gives the regret bound of $\mathcal{O}(K)$ for any arbitrary learning algorithm. To achieve a regret-free algorithm, we define an episode-specific deadline that categorizes learning episodes as fast or slow. We prove that the number of slow episodes exceeding the deadline grows sublinearly. The deadline is set such that the regret from fast episodes also grows sublinearly. Thus, the total regret of our algorithm increases sublinearly, making

it regret-free. We utilize a reset mechanism that enables a systematic trade-off between exploration and exploitation, and can be applied to non-communicating MDPs. This is unlike the previous work by Tarbouriech et al. (2020), which is limited to strictly positive cost structures and properties with satisfaction probability one, and surpasses the limitations of UCRL2-like algorithms, which require the underlying MDP to be communicating. The ability to handle non-communicating MDPs is key since the product of an MDP and an automaton for the temporal specification makes in general the product MDP to be non-communicating.

We have implemented our algorithm with empirical evaluation on reach-avoid specifications in a gridworld environment reported in App. C. In particular, we compare the performance and convergence of our algorithm to the sample complexity and convergence of a PAC-MDP algorithm for LTL (Perez et al., 2024), and show that our algorithm convergences faster within a smaller number of episodes.

## 2. Related Work

We discuss existing results in four related domains.

**Reinforcement Learning for LTL Specifications Without Guarantees:** In recent years, substantial research has focused on the policy synthesis problem for systems modeled as finite MDPs with unknown transition probabilities, where the objective is to satisfy tasks specified by LTL formulas. Early results focused only on finite-horizon LTL specifications. Icarte et al. (2018) introduced reward machines, which use finite state automata to encode finite-horizon specifications along with specialized (deep) Q-learning algorithms. Camacho et al. (2019) later formalized the automatic derivation of reward machines for finite-horizon LTL specifications. The development of good-for-MDP automata, such as limit deterministic Büchi automata (LDBA), for representing LTL formulas has led to significant advances in reinforcement learning for the full class of LTL specifications, including the infinite-horizon ones (Sickert et al., 2016; Hahn et al., 2020; Kazemi et al., 2022; Hasanbeig et al., 2023). Typically, one has to first translate the given LTL formula into an appropriate automaton, such as an LDBA, and then compute the product of this automaton with the MDP to formulate the final (discounted) learning problem. This formulation ensures that, with a sufficiently large discount factor (depending on system dynamics and the specification), applying standard RL algorithms will lead the policy to converge asymptotically to the optimal one (Kazemi et al., 2022; Bozkurt et al., 2021; Oura et al., 2020). Translation of the LTL formula to an average objective for RL using the automaton construction is also studied by Kazemi et al. (2025). However, these methods do not provide a finite-time performance, and the required discount factor is not known in advance.

**Reinforcement Learning for LTL Specifications With Guarantees:** The two most popular metrics for evaluating the performance of learning algorithms are probably approximately correct (PAC) and regret bounds. Fu & Topcu (2014) have provided a PAC learning algorithm for synthesizing policies to satisfy finite-horizon LTL specifications over finite MDPs with unknown transition probabilities. However, the sample complexity of the algorithm scales explicitly with the horizon length, rendering it unsuitable for infinite-horizon LTL specifications. Subsequently, a surprising negative result was established: full LTL is not PAC-learnable (Yang et al., 2022; Alur et al., 2022). A closer analysis reveals that this result primarily stems from the assumption of an *unknown minimum transition probability*. By instead assuming a known minimum transition probability, Voloshin et al. (2022) proposed a PAC-learnable policy synthesis method that relies on access to a generative model capable of sampling from any arbitrary state-action pair. In many realistic scenarios, this is impractical since initializing the MDP at arbitrary states is not feasible. Our proposed algorithm does not require access to a generative model. Recently, Perez et al. (2024) have also proposed another PAC learning algorithm that does not rely on a generative model. However, there is still no regret-free online algorithm for policy synthesis to satisfy LTL specifications.

**Regret-Free Reinforcement Learning for Communicating MDPs:** UCRL and UCRL2 are well-known regret-free learning algorithms developed for communicating MDPs (Auer et al., 2008; Auer & Ortner, 2007). The definition of regret in the context of communicating MDPs is particularly well-suited to our objective: it is evaluated over an infinite sequence of states without discounting future observations, which aligns closely with the requirements of infinite-horizon LTL specifications. A key advantage of learning in communicating MDPs is that it can proceed indefinitely without requiring environment resets. However, in the setting we consider, even if the underlying MDP is communicating, its product with the automaton representing the specification may result in a non-communicating MDP (Kazemi et al., 2022). Fruit et al. (2018) address this by proposing a regret-free algorithm for non-communicating MDPs where the initial state lies within a non-transient subset of states. In contrast, our setting assumes a fixed initial state from a transient subset, rendering algorithms like UCRL2–which are tailored for communicating MDPs– inapplicable.

**Regret-Free Reinforcement Learning for Non-Communicating MDPs:** Goal-oriented reinforcement learning is a key class of problems in RL, often formulated as a *shortest path problem (SPP)* for MDPs with unknown transition probabilities. Recent theoretical results include the works by Tarbouriech et al. (2020) and Rosenberg et al. (2020). In particular, the online learning algorithm

proposed by Tarbouriech et al. (2020) provides sublinear regret bounds for the accumulated cost in MDPs, assuming that (1) there exists a proper policy under which the system reaches the goal with probability one, and that (2) all costs are positive. The authors argue that these two restrictions can be relaxed by assuming the knowledge of an upper bound on the accumulated cost and by perturbing the costs. However, none of these workarounds can be applied to our setting, as we are interested in computing the regret bounds with respect to the satisfaction probability under an optimal policy. Nevertheless, relaxing the aforementioned restrictions results in the algorithm by Tarbouriech et al. (2020) giving a regret bound of $\mathcal{O}(K^{2/3})$, which is strictly larger than our regret bound of $\mathcal{O}(K^{1/2})$, with $K$ being the number of episodes.

## 3. Preliminaries

**Notation:** For a matrix $X \in \mathbb{R}^{m \times n}$, we define the infinity norm of the matrix by $\|X\|_\infty :=$ $\max_{1 \le i \le m} \sum_{j=1}^n |X(i,j)|$. For a vector $x \in \mathbb{R}^n$, we define the $\ell_1$ and $\ell_\infty$ norms as $\|x\|_1 := \sum_{i=1}^n |x(i)|$ and $\|x\|_\infty := \max_{1 \le i \le n} |x(i)|$, respectively. Given two integers $a, b$ with $a \le b$, we denote the set of integers $\{a, a+1, \ldots, b\}$ by $[a;b]$. For a set $S$, we denote by $S^+$ the set of all non-empty finite sequences from the elements of $S$. Let $\emptyset$ denote the empty set. For a sequence $\sigma \in S^+$, we define the corresponding last element by $last(\sigma)$. The set of positive integers is denoted by $\mathbb{N}$ and the set of non-negative integers by $\mathbb{N}_0$. We use $|A|$ to denote the cardinality of a set $A$ (i.e., the number of its elements).

**MDPs:** For a set $X$, we write $\Delta(X)$ for the set of probability distributions over $X$. Let $AP$ be a fixed set of atomic propositions. A labeled MDP $M = (S, A, T, s_{\mathsf{init}}, L, AP)$ consists of a finite set $S$ of states, a finite set $A$ of actions, a transition function $T: S \times A \to \Delta(S)$, an initial state $s_{\mathsf{init}} \in S$, and a *labeling function* $L: S \to 2^{AP}$ that maps states to subsets of atomic propositions $AP$. The *underlying graph* of an MDP $M$ is defined as $\chi(M) = (S, A, E)$, where $(s, a, s') \in E$ if and only if $T(s'|s, a) > 0$.

An iMDP is defined similarly to an MDP, but instead of a single transition probability, each transition is specified by an interval of possible probabilities. Formally, an iMDP is a tuple $\mathcal{M} = (S, A, \mathcal{T}, s_{\mathsf{init}}, L, AP)$, where $\mathcal{T}: S \times A \to I(S)$ is an *interval transition function*, and $I(S)$ denotes the set of mappings that assign to each $s' \in S$ a closed interval $[l(s'), u(s')] \subseteq [0, 1]$, such that $\sum_{s' \in S} l(s') \le 1 \le \sum_{s' \in S} u(s')$.

A *policy* for the MDP is a mapping $\pi : S^+ \to \Delta(A)$ that gives a probability distribution $\pi(\sigma) \in \Delta(A)$ for selecting the next action depending on $\sigma \in S^+$, which is the nonempty finite sequence of states representing the past

history. A policy $\pi$ is *memoryless* if $\pi(\sigma) = \pi(s)$ for all $\sigma \in S^+$ ending in $s$ and for all $s \in S$. Let $\Pi$ denote the set of all deterministic *positional* policies over $M$, that is the set of functions $\pi \colon S \to A$. By fixing a policy $\pi \in \Pi$, the MDP $M$ reduces to a *Markov chain* $\mathcal{C} = (S, P, s_{\text{init}}, L, AP)$, where $P \colon S \to \Delta(S)$ is formed by composing the policy $\pi$ with the transition function $T$.

A key concept for checking satisfaction of specifications is *maximal end components* (MECs) of the MDP (Baier & Katoen, 2008). These components are sub-MDPs that are probabilistically closed, meaning that (1) there exists a positional policy under which the probability of reaching any state from any other state in the MEC is equal to one, and (2) the MEC cannot be exited under any positional policy. An MDP is said to be *communicating* if it consists of a single MEC that includes all states.

**Linear Temporal Logic:** We consider specifications in Linear Temporal Logic (LTL) (Baier & Katoen, 2008). Formulas in LTL are constructed inductively over a set of atomic propositions $AP$ according to the syntax

$$\psi := p \mid \neg\psi \mid \psi_1 \wedge \psi_2 \mid \bigcirc\psi \mid \psi_1 \cup \psi_2,$$

where $p \in AP$. The semantics of LTL is defined on infinite sequences of elements from $2^{AP}$. Let $\sigma = \sigma_0, \sigma_1, \ldots$ be an infinite sequence of elements from $2^{AP}$ and define $\sigma[i] = \sigma_i, \sigma_{i+1}, \ldots$ for any $i \in \mathbb{N}_0$. Then the satisfaction relation between $\sigma$ and a property $\psi$, expressed in LTL, is denoted by $\sigma \models \psi$. We have $\sigma \models p$ if $p \in \sigma_0$. Furthermore, $\sigma \models \neg\psi$ if $\sigma \not\models \psi$ and $\sigma \models \psi_1 \wedge \psi_2$ if $\sigma \models \psi_1$ and $\sigma \models \psi_2$. For *next* operator, $\sigma \models \bigcirc\psi$ holds if $\sigma[1] \models \psi$. The *until* operator $\sigma \models \psi_1 \cup \psi_2$ holds if $\exists i \in \mathbb{N}_0 : \sigma[i] \models \psi_2$, and $\forall j \in \mathbb{N}_0, j < i, \sigma[j] \models \psi_1$. We define derived operators such as disjunction ($\vee$), eventually ($\Diamond$), and globally ($\square$) in the usual way.

**Maximum Probability of Satisfaction:** Take an MDP $M$ and an LTL specification $\varphi$. A path $s_0, s_1, \ldots$ of $M$ with $s_0 = s_{\text{init}}$ satisfies $\varphi$ if $L(s_0), L(s_1), \ldots \models \varphi$. For every policy $\pi$, the set of paths starting at $s_{\text{init}}$ that satisfy $\varphi$ is measurable. Thus, we can define $\mathbb{P}^\pi_{s_{\text{init}}}(\varphi)$, the probability that $M$ under policy $\pi$ satisfies $\varphi$, where with abuse of notation, we write $\varphi$ for the set of paths satisfying $\varphi$. Define the optimal probability of satisfying $\varphi$ with $v^*(s_{\text{init}}) = \sup_\pi \mathbb{P}^\pi_{s_{\text{init}}}(\varphi)$, and let $\pi^*$ denote the optimal policy, i.e., $v^*(s_{\text{init}}) = \mathbb{P}^{\pi^*}_{s_{\text{init}}}(\varphi)$. We assume that the specification is satisfiable with positive probability, i.e., $v^*(s_{\text{init}}) > 0$. This is without loss of generality since *any* learning algorithm for the case $v^*(s_{\text{init}}) = 0$ has a zero regret.

**Regret Analysis:** Our aim is to learn the optimal policy $\pi^*$ that maximizes $\mathbb{P}^\pi_{s_{\text{init}}}(\varphi)$. Learning takes place over consecutive episodes. In episode $k$, we learn a policy $\pi_k$, and write $v_k(s_{\text{init}}) = \mathbb{P}^{\pi_k}_{s_{\text{init}}}(\varphi)$, which is the probability of satisfying the specification $\varphi$ under the policy $\pi_k$.

We define the *regret* of the learning algorithm as

$$R(K) := \sum_{k=1}^K (v^*(s_{\text{init}}) - v_k(s_{\text{init}})), \qquad (1)$$

where $K$ is the number of episodes, and the *normalized regret* as

$$R_a(K) = \frac{R(K)}{K}. \qquad (2)$$

A learning algorithm is called *regret-free* if its regret $R(K)$ grows sublinearly with respect to the number of episodes $K$, i.e., if $R_a(K) \to 0$ as $K \to \infty$. A regret-free algorithm can achieve arbitrary small values of normalized regret. Suppose we fix a threshold $\varepsilon \in (0, 1)$ and terminate the learning algorithm once the corresponding normalized regret goes below $\varepsilon$. We can consider the *smallest* number of episodes $k^* \in \mathbb{N}$ after which $R_a(k^*) < \varepsilon$ with confidence $1 - \delta$ as a complexity metric for the proposed learning algorithm with parameters $\delta, \varepsilon \in (0, 1)$.

## 4. Regret-Free Learning for Until Formulas

We first consider the special case of regret-free learning for MDPs with unknown (but fixed) transition function and *until* formulas that correspond to *reach-avoid* specifications.

For two distinct atomic propositions Goal and Avoid, we consider the *until* formula (also called a *reach-avoid* formula) of the form $\varphi = \neg\text{Avoid } \cup \text{ Goal}$. Let $M_\varphi = (S, A, T, s_{\text{init}}, L, \{\text{Goal}, \text{Avoid}\})$ be the MDP obtained by making states satisfying Goal and Avoid absorbing. We use $G$ and $B$ to denote the sets of states which satisfy the atomic propositions Goal and Avoid, respectively. For notational convenience, we write $M$ instead of $M_\varphi$ for the remainder of this section.

---

*Problem* 1. [Regret-Free Learning for Until Formulas]
Given an MDP $M$ with an unknown transition function, an until formula $\varphi = \neg\text{Avoid } \cup \text{ Goal}$, a confidence parameter $\delta \in (0, 1)$, and a minimum transition probability $p_{min} \in (0, 1)$, find an online learning algorithm such that with confidence at least $(1 - \delta)$, the regret defined by Eq. (1) grows sublinearly with the number of episodes $K$.

---

### 4.1. Methodology

Alg. 1 shows our learning algorithm. It uses the paradigm of *optimism in the face of uncertainty*. Learning takes place over consecutive *environmental episodes*. Each episode is a *finite* sequence $s_1, a_1, s_2, a_2, \ldots s_H$ that starts from the initial state of $M$, i.e., $s_1 = s_{\text{init}}$, and ends if either an absorbing state in $G$ is reached, meaning that $s_H \in G$, or an *episode-specific deadline* is reached.

**Algorithm 1** Regret-free algorithm for until formulas (ZeroReg)

**Input:** State and action sets $S$ and $A$, initial state $s_{\text{init}}$, sets $G$ and $B$, confidence parameter $\delta \in (0,1)$, minimum transition probability $p_{min}$
Initialization: Set $t = 1$, $s_1 = s_{\text{init}}$
**for** episodes $k = 1, 2, \ldots$ **do**
  **Construct iMDP $\mathcal{M}_k$:**
  $t_k \leftarrow t$
  Set $N_k(s,a) \coloneqq |\{t < t_k : s_t = s, a_t = a\}|$
  For all $s, s' \in S$ and $a \in A$ compute the empirical transition function

$$\hat{T}_k(s'|s,a) \coloneqq \frac{|\{t < t_k : s_t = s, a_t = a, s_{t+1} = s'\}|}{\max\{1, N_k(s,a)\}}$$

  Define the iMDP $\mathcal{M}_k = (S, A, \mathcal{T}_k, s_{\text{init}}, L, AP)$ with the set of transition functions $\mathcal{T}_k$ satisfying Eq. (3)
  **Compute policy $\tilde{\pi}_k$:**
  Use Alg. 2 to find an optimistic MDP $\tilde{M}_k$ and a policy $\tilde{\pi}_k$, i.e., $(\tilde{M}_k, \tilde{\pi}_k) = \text{EVI}(S, A, G, B, \mathcal{T}_k, t_k, p_{min})$
  **Execute policy $\tilde{\pi}_k$:**
  Compute the deadline $H_k$ using Eq. (7)
  **while** $s_t \notin G$ and $(t - t_k) \leq H_k$ **do**
    **if** $s_t \notin B$ **then**
      Observe the next state $s_{t+1}$ by executing $\tilde{\pi}_k(s_t)$
    **else**
      $s_{t+1} \leftarrow s_{\text{init}}$
    **end if**
    $t \leftarrow t + 1$
  **end while**
**end for**

---

**Algorithm 2** Extended value iteration (EVI)

**Input:** State and action sets $S$ and $A$, sets $G$ and $B$, set of plausible transition functions $\mathcal{T}_k$, start time of the $k^{th}$ episode $t_k$, minimum transition probability $p_{min}$
Set $l = 0$, $\tilde{\mu}_0(s) = 0$ for $s \notin G$ and $\tilde{\mu}_0(s) = 1$ for $s \in G$
**repeat**
  $l \leftarrow l + 1$
  **for** $s \in S$ **do**
    **for** $a \in A$ **do**
      $\tilde{T}_k(.|s,a) = \text{InnerMax}(s, a, \mathcal{T}_k, \tilde{\mu}_{l-1})$
    **end for**
    $\tilde{\pi}_k(s) = \arg\max_{a \in A} \sum_{s' \in S} \tilde{T}_k(s'|s,a)\tilde{\mu}_{l-1}(s')$
    $\tilde{\mu}_l(s) = \sum_{s' \in S} \tilde{T}_k(s'|s, \tilde{\pi}_k(s))\tilde{\mu}_{l-1}(s')$
  **end for**
**until** $\|\tilde{\mu}_l - \tilde{\mu}_{l-1}\|_\infty < \min(\frac{1}{2t_k}, p_{min}^{|S|})$
Construct the MDP $\tilde{M}_k = (S, A, \tilde{T}_k, s_{\text{init}}, L, AP)$
**Results:** $(\tilde{M}_k, \tilde{\pi}_k) = \text{EVI}(S, A, G, B, \mathcal{T}_k, t_k, p_{min})$

---

Each episode proceeds as follows: (1) construct an iMDP from observations such that it contains the true MDP with high confidence; (2) identify an optimistic MDP and compute an *optimistic* policy; and (3) execute the policy and collect data until termination. We describe each of these steps in detail below.

**Constructing an iMDP from observations:** The main objective of this step is to use the collected data–namely, the visited state-action pairs $(s_t, a_t)$ and the corresponding transitions $(s_t, a_t, s_{t+1})$–to compute an empirical transition function and construct an iMDP that contains the true MDP with high confidence.

Let $\delta \in (0,1)$ be a given confidence parameter, $t_k$ be the time point at which $k^{th}$ episode begins, and $N_k(s,a)$ denote the number of times the state-action pair $(s,a)$ has been visited before the start of the $k^{th}$ episode. Let $\hat{T}_k$ and $\mathcal{M}_k$ denote the empirical transition function and the set of statistically plausible MDPs, respectively, both computed using the observations before the start of the $k^{th}$ episode. In

particular, we define $\mathcal{M}_k = (S, A, \mathcal{T}_k, s_{\text{init}}, L)$ as the iMDP with interval transition function $\mathcal{T}_k$, such that, with probability at least $(1 - \delta/3)$, every transition function $F \in \mathcal{T}_k$ satisfies

$$\|F(.|s,a) - \hat{T}_k(.|s,a)\|_1 \leq \beta_k(s,a), \qquad (3)$$

where

$$\beta_k(s,a) \coloneqq \sqrt{\frac{8|S| \log(2|A|N_k^+(s,a)/\delta)}{N_k^+(s,a)}}, \qquad (4)$$

and $N_k^+(s,a) \coloneqq \max(1, N_k(s,a))$. Intuitively, we pick the confidence bound $\delta$ on the right hand side of Eq. (3), such that the corresponding inequality holds with high confidence. More concretely, we have the following result.

**Lemma 4.1.** *Let* $\mathcal{E} \coloneqq \bigcap_{k=1}^\infty \{M \in \mathcal{M}_k\}$. *Then* $\mathbb{P}(\mathcal{E}) \geq 1 - \delta/3$.

The above lemma states that the true MDP $M$ lies within the iMDP $\mathcal{M}_k$, constructed from observations, with high probability. Next, we describe the policy synthesis procedure based on $\mathcal{M}_k$.

**Computing optimistic policy:** Given an iMDP constructed from the observations in the previous step, an *optimistic MDP* is a specific MDP selected from the uncertainty set– defined by interval transition probabilities–that maximizes the probability of reaching $G$ while avoiding $B$. This optimistic MDP is then used to derive an *optimistic policy* that guides exploration by assuming the most favorable dynamics permitted within the uncertainty bounds. In every episode $k \in \mathbb{N}$, we use a modified version of extended value iteration (EVI) to compute theoptimistic MDP $\tilde{M}_k \in \mathcal{M}_k$ and the optimistic policy $\tilde{\pi}_k$. Alg. 2 illustrates our EVI algorithm. We initialize the vector $\tilde{\mu}_0 \in [0,1]^{|S|}$ by assigning

**Algorithm 3** Computing optimistic MDP in Eq. (5) (InnerMax)

---

**Inputs:** State-action pair $(s, a)$, interval transition function $\mathcal{T}_k$, and value vector $\tilde{\mu}_l$

Compute $\hat{T}_k(.|s, a)$ and $d(s, a)$ as the center and $\ell_1$ radius of $\mathcal{T}_k(.|s, a)$, respectively

Sort the states according to their value in $\tilde{\mu}_l$, and get $S' = \{s'_1, \ldots, s'_n\}$ with $\tilde{\mu}_l(s'_1) \geq \tilde{\mu}_l(s'_2) \geq \cdots \geq \tilde{\mu}_l(s'_n)$

**if** $j = 1$ **then**
  $\hat{p}(s'_j) = \hat{T}_k(s, a, s'_j) + d(s, a)/2$
**else**
  $\hat{p}(s'_j) = \hat{T}_k(s, a, s'_j)$
**end if**
$j \leftarrow n$
**while** $\sum_{s'_q \in S'} \hat{p}(s'_q) > 1$ **do**
  $\hat{p}(s'_j) = \max\{0, 1 - \sum_{s'_q \in S' \setminus \{s'_j\}} \hat{p}(s'_q)\}$
  $j \leftarrow j - 1$
**end while**
Set $\tilde{T}_k(s, a, s'_j) = \hat{p}(s'_j)$ for every $s'_j \in S'$.
**Results:** $\tilde{T}_k(.|s, a) = \mathrm{InnerMax}(s, a, \mathcal{T}_k, \tilde{\mu}_l)$

---

1 to states in $G$ and 0 to all other states. At the $l^{th}$ iteration, we update the value vector $\mu_l$ by applying the following Bellman operator:

$$\tilde{\mu}_l(s) = \tilde{\mathcal{L}}_k \tilde{\mu}_{l-1} :=$$
$$\max_{a \in A} \max_{F \in \mathcal{T}_k} \sum_{s' \in S} F(s'|s, a)\tilde{\mu}_{l-1}(s'), \qquad (5)$$

where the inner maximization is computed by running Alg. 3.

We note that termination of Alg. 2 requires convergence of the value vector, i.e., $\|\tilde{\mu}_l - \tilde{\mu}_{l-1}\|_\infty < \min(\frac{1}{2t_k}, p_{min}^{|S|})$, where $l \in \mathbb{N}_0$ denotes the iteration number of the EVI algorithm. This condition is guaranteed due to the contraction property of the Bellman operator defined in Eq. (5) (see Sec. 3.3 in (Bertsekas, 2012)). Once EVI algorithm is terminated, setting $\tilde{v}_k = \tilde{\mu}_l$, it follows that for every $s \in S$,

$$\tilde{v}_k(s) + \min(\frac{1}{2t_k}, p_{min}^{|S|}) \geq v^*(s), \qquad (6)$$

where $v^*$ and $\tilde{v}_k$ denote the vectors containing probabilities of reaching $G$, when policies $\tilde{\pi}_k$ and $\pi^*$ are followed on MDPs $\tilde{M}_k$ and $M$, respectively (see Thm. 7 in (Auer et al., 2008), ).

**Executing the policy and collecting data:** Once a policy is computed, it will be executed throughout the episode for a specified duration, referred to as the *deadline*. After constructing $\tilde{M}_k$ and computing the corresponding policy $\tilde{\pi}_k$ for episode $k$, we determine the deadline $H_k$, which is defined as

$$H_k = \min\{n > 1 \mid \|\tilde{Q}_k^n\|_\infty \leq k^{-\frac{1}{q}}\}, \qquad (7)$$

where $q > 1$ denotes an integer, and $\tilde{Q}_k \in \mathbb{R}^{(|S|-|G|) \times (|S|-|G|)}$ is a *substochastic* matrix defined over the set of states $S \setminus G$ as follows:

$$\tilde{Q}_k(s, s') = \begin{cases} \tilde{P}_k(s, s') & s \notin B \text{ and } s' \in S \setminus G \\ 1 & s \in B \text{ and } s' = s_{\mathsf{init}} \\ 0 & s \in B \text{ and } s' \neq s_{\mathsf{init}}. \end{cases} \qquad (8)$$

Note that we intentionally set the transition probability from $B$ to $s_{\mathsf{init}}$ to 1 to account for the reset mechanism. Additionally, since $\tilde{Q}_k$ is substochastic, $\lim_{n \to \infty} \|\tilde{Q}_k^n\|_\infty = 0$, ensuring that $H_k$ is finite for every episode $k \in \mathbb{N}$.

As described in Alg. 1, the $k^{th}$ episode terminates either upon reaching a state in $G$ or when its duration reaches the deadline $H_k$. During the episode, we record the visited state-action pairs $(s_t, a_t)$ along with the resulting transitions $(s_t, a_t, s_{t+1})$. These transitions are then used to construct the iMDP at the beginning of the next episode, as previously explained.

**Regret bound analysis:** The following theorem states that Alg. 1 is regret free.

**Theorem 4.2.** *With probability at least $(1 - 2\delta)$, Alg. 1 has regret*

$$R(K) = 8|S|\sqrt{8|A|K\alpha_K \log\left(\frac{2|A|K\alpha_K}{\delta}\right)}$$
$$+ 2\sqrt{2K\alpha_K \log\left(\frac{6(K\alpha_K)^2}{\delta}\right)}$$
$$+ \frac{1}{2}\alpha_K(1 + \log(K\alpha_K))$$
$$+ \sqrt{2K\alpha_K \log\left(\frac{1}{\delta}\right)}$$
$$+ 2\sqrt{K} + 2\sqrt{2K\alpha_K \log\left(\frac{2(K\alpha_K)^2}{\delta}\right)}, \qquad (9)$$

*where $\alpha_K := \max_{1 \leq k \leq K} H_k$.*

*Remark* 4.3. The quantity $\alpha_K$ grows at most logarithmically with respect to the number of episodes $K$ (see Lem. 4.4). Therefore, the regret bound in Eq. (9) grows sublinearly with respect to the number of episodes.

### 4.2. Proof Sketch for Thm. 4.2

Every episode starts at $s_{\mathsf{init}}$ and ends by either (i) exceeding the deadline, corresponding to *slow* episodes, or (ii) by reaching one of the MECs in $G$, corresponding to *fast* episodes. It is important to note that every visit to states within MECs in $B$ triggers an artificial reset action, which sets $s_{\mathsf{init}}$ as the next immediate state.

To bound the total accumulated regret $R(K)$, we define $R(K) = \sum_{k=1}^{K} \Delta_k$, where $\Delta_k = v^*(s_{\text{init}}) - v_k(s_{\text{init}})$. Our analysis partitions episodes into *slow* and *fast*, corresponding to episodic regrets $\Delta_k^{\text{s}}$ and $\Delta_k^{\text{f}}$, respectively. Note that for a fast episode, $\Delta_k^{\text{s}} = 0$, and for a slow episode, $\Delta_k^{\text{f}} = 0$.

For the slow episodes, we use the obvious upper bound

$$\Delta_k^{\text{s}} \leq 1.$$

For the fast episodes, since it is possible that a run ends in one of MECs in $B$ before reaching $G$, we need to define a reset transition which takes the states in $B$ to $s_{\text{init}}$. Therefore, every episode $k$ can be broken to $I_k \in \mathbb{N}$ *intervals* such that the first $I_k - 1$ intervals start from $s_{\text{init}}$ and end at $B$, and the $I_k^{th}$ interval starts from $s_{\text{init}}$ and ends at $G$.

We denote the $i^{\text{th}}$ interval of the $k^{\text{th}}$ episode—during which the policy $\tilde{\pi}_k$ is applied—by $\rho_{k,i}$, and define the corresponding value as

$$v_{k,i}(s_{\text{init}}) = \begin{cases} 1 & \text{if } last(\rho_{k,i}) \in G \\ 0 & \text{if } last(\rho_{k,i}) \in B. \end{cases} \quad (10)$$

We use the fact that $v^*(s_{\text{init}}) - v_k(s_{\text{init}}) \leq I_k(v^*(s_{\text{init}}) - v_k(s_{\text{init}}))$ (since $I_k \geq 1$) and define

$$\Delta_k^{\text{f}} \leq I_k(v^*(s_{\text{init}}) - v_k(s_{\text{init}}))$$
$$= \sum_{i=1}^{I_k} v^*(s_{\text{init}}) - v_{k,i}(s_{\text{init}})$$
$$+ \sum_{i=1}^{I_k} v_{k,i}(s_{\text{init}}) - v_k(s_{\text{init}}).$$

To obtain an upper bound on $\Delta_k^{\text{f}}$, we overapproximate the difference $v^*(s_{\text{init}}) - v_k(s_{\text{init}})$ in every episode, since the exact values of $v^*(s_{\text{init}})$ and $v_k(s_{\text{init}})$ are not known. This is achieved by leveraging ideas from upper confidence bound algorithms, such as UCRL2, and by relating $v_k(s_{\text{init}})$ to the cumulative sum $\sum_{i=1}^{I_k} v_{k,i}(s_{\text{init}})$.

We define the decomposed regret terms

$$\Delta_k^{\text{f1}} = \sum_{i=1}^{I_k} v^*(s_{\text{init}}) - v_{k,i}(s_{\text{init}}),$$

and

$$\Delta_k^{\text{f2}} = \sum_{i=1}^{I_k} v_{k,i}(s_{\text{init}}) - v_k(s_{\text{init}}).$$

The proof sketch for our regret analysis proceeds as follows. We first show that the sum $\sum_{k=1}^{K} \Delta_k^{\text{f}}$ grows logarithmically with the number of episodes (Lems. 4.4 to 4.6). To this

end, we decompose the sum into two parts, $\sum_{k=1}^{K} \Delta_k^{\text{f1}}$ and $\sum_{k=1}^{K} \Delta_k^{\text{f2}}$, and show that both grow sublinearly in $K$. To bound $\sum_{k=1}^{K} \Delta_k^{\text{f1}}$, we prove that (1) $\alpha_K$ grows sublinearly with $K$ (Lem. 4.4), and (2) $\sum_{k=1}^{K} \Delta_k^{\text{f1}}$ grows sublinearly with $K$ and linearly with the maximum episode length $\alpha_K$ (Lem. 4.5). To complete the bound on $\sum_{k=1}^{K} \Delta_k^{\text{f}}$, we show that $\sum_{k=1}^{K} \Delta_k^{\text{f2}}$ also grows sublinearly (Lem. 4.6). Next, we show that the sum over the slow episodes, $\sum_{k=1}^{K} \Delta_k^{\text{s}}$, is sublinear in $K$, since (1) the number of slow episodes grows only sublinearly with $K$ (Lem. 4.7), and (2) each $\Delta_k^{\text{s}} \leq 1$ by definition. Combining all these bounds, we conclude that the total regret satisfies $R(K) = \mathcal{O}(\sqrt{K})$.

The following lemma provides an upper bound on the maximum episode length, which grows logarithmically with the episode number.

**Lemma 4.4.** *With probability at least $(1 - \delta/6)$, we have*

$$\alpha_K \leq \left\lceil 3\Lambda \log(2\sqrt{K}) \right\rceil, \quad (11)$$

*where $\Lambda = |S| \frac{\log(\delta/6)}{\log(1 - p_{min}^{|S|})}$.*

Now, we proceed by showing why $\sum_{k=1}^{K} \Delta_k^{\text{f1}}$ grows sublinearly with $K$.

**Lemma 4.5.** *With probability at least $(1 - 5\delta/6)$, we have*

$$\sum_{k=1}^{K} \Delta_k^{\text{f1}} \leq 4|S|\sqrt{8|A|K\alpha_K \log\left(\frac{2|A|K\alpha_K}{\delta}\right)}$$
$$+ 2\sqrt{2K\alpha_K \log\left(\frac{6(K\alpha_K)^2}{\delta}\right)}$$
$$+ \frac{1}{2}\alpha_K(1 + \log(K\alpha_K)). \quad (12)$$

In order to prove the sublinear bound over $\sum_{k=1}^{K} \Delta_k^{\text{f2}}$, we make use of the Azuma-Hoeffding inequality (see Lem. B.2). The following lemma provides a sublinear bound over the sum of $\Delta_k^{\text{f2}}$.

**Lemma 4.6.** *With probability at least $(1 - \delta/6)$, we have*

$$\sum_{k=1}^{K} \Delta_k^{\text{f2}} \leq \sqrt{2K\alpha_K \log\frac{1}{\delta}}. \quad (13)$$

Note that knowing that $\alpha_K$ grows logarithmically with $K$ (Lem. 4.4), we have that $\sum_{k=1}^{K} \Delta_k^{\text{f1}}$ and $\sum_{k=1}^{K} \Delta_k^{\text{f2}}$, and therefore $\sum_{k=1}^{K} \Delta_k^{\text{f}}$ grow sublinearly with $K$.

The next lemma states a bound over the accumulated regret associated with the slow episodes.

**Lemma 4.7.** *With probability at least $(1 - \delta)$, we have*

$$\sum_{k=1}^{K} \Delta_k^{\mathsf{s}} \leq 2\sqrt{K} + 2\sqrt{2K\alpha_K \log\left(\frac{2(K\alpha_K)^2}{\delta}\right)}$$

$$+ 4|S|\sqrt{8|A|K\alpha_K \log\left(\frac{2|A|K\alpha_K}{\delta}\right)}. \quad (14)$$

## 5. Regret-Free Learning for LTL

In this section, we study the policy synthesis problem for MDPs with an unknown transition function against LTL specifications.

> *Problem* 2. [Regret-Free Learning for General LTL Formulas] Given an MDP $M$ with an unknown transition function, minimum transition probability $p_{min} \in (0, 1)$, an LTL specification $\varphi$, and a confidence parameter $\delta \in (0, 1)$, find an online learning algorithm such that with confidence at least $(1 - \delta)$ the resulting regret defined by Eq. (1) grows sublinearly with respect to the number of episodes $K$.

We transform the policy synthesis problem for general LTL specifications into a synthesis problem for an until (reach-avoid) formula, for which one can use the regret-free online algorithm proposed in Sec. 4.

Given an LTL formula $\varphi$, one can construct a *deterministic Rabin automaton* (DRA) whose language corresponds to all infinite sequences of system behaviors satisfying $\varphi$. This automaton, denoted by $\mathfrak{A}_\varphi = (Q, \Sigma, \gamma, q_{\text{init}}, F)$, consists of a finite set of states $Q$, a finite alphabet $\Sigma = 2^{AP}$, a transition function $\gamma \colon Q \times \Sigma \to Q$, an initial state $q_{\text{init}}$, and an accepting condition $F = \{(J_i, K_i) \mid i = 1, \dots, m\}$, consisting of subsets $J_i$ and $K_i$ of $Q$. An infinite sequence $\sigma$ is accepted by $\mathfrak{A}_\varphi$ if there exists at least one pair $(J, K) \in F$ such that $\inf(\sigma) \cap J = \emptyset$ and $\inf(\sigma) \cap K \neq \emptyset$, where $\inf(\sigma)$ is the set of states that appear infinitely often in $\sigma$.

For a given LTL specification and an MDP, the corresponding optimal policy, in general, belongs to the class of (deterministic) non-positional policies, that are mappings from the finite *paths* over the MDP into the set of actions. One can restrict the set of policies to positional policies by constructing the corresponding *product MDP*, obtained as the product between the MDP $M$ and the automaton $\mathfrak{A}_\varphi$. Given an MDP $M = (S, A, T, s_{\text{init}}, L, AP)$ and a DRA $\mathfrak{A}_\varphi = (Q, \Sigma, \gamma, q_{\text{init}}, F)$, we denote the product MDP by the tuple $M_\varphi = (S^\times, A^\times, T^\times, s_{\text{init}}^\times, L^\times, AP^\times)$, where $S^\times = S \times Q$, $A^\times = A$, $s_{\text{init}}^\times = (s_{\text{init}}, q_{\text{init}})$, $AP^\times = AP$, $L^\times \colon (s, q) \mapsto L(s)$ for every $(s, q) \in S \times Q$, and $T^\times \colon S^\times \to \Delta(S^\times)$ taking the form

$$T^\times((s, q), a, (s', q')) = \begin{cases} T(s, a, s') & q' = \gamma(q, L(s')) \\ 0 & \text{otherwise.} \end{cases}$$

In App. A, we show a polynomial algorithm for learning the underlying graph of the MDP $M$, i.e., $\chi(M)$, using the knowledge of $p_{min}$ (see Alg. 5). Knowledge of $\chi(M)$ directly gives the graph for the product MDP $M_\varphi$, that is $\chi(M_\varphi) = (S^\times, A^\times, E^\times)$, where $((s, q), a, (s', q')) \in E^\times$ if and only if

$$((s, q), a, (s', q')) \in E^\times \Leftrightarrow$$
$$(s, a, s') \in E \text{ and } q' \in \gamma(q, L(s')). \quad (15)$$

Let $D \subset S$ denote the set of states of a specific MEC $C$ within $M_\varphi$. We say $C$ is an accepting maximal end component (AMECs) within $M_\varphi$ if and only if

$$D \cap \bigcup_{i=1}^{m} S \times K_i \neq \emptyset, \text{ and } D \cap \bigcup_{i=1}^{m} S \times J_i = \emptyset. \quad (16)$$

We denote the set of all states corresponding to the accepting and non-accepting MECs within $M_\varphi$ by $G^\times$ and $B^\times$, respectively.

Alg. 4 outlines our proposed online regret-free algorithm for solving the policy synthesis problem against LTL specifications. First, we compute a DRA $\mathfrak{A}_\varphi$ that accepts $\varphi$. We can run Alg. 5 and use Eq. (15) to get the underlying graph of $M$ and $M_\varphi$, respectively. Once we know the graph of $M_\varphi$, we can use Algorithm 47 from (Baier & Katoen, 2008) to characterize all of the MECs in $M_\varphi$. Next, we include the states within accepting and non-accepting MECs into $G^\times$ and $B^\times$, resepectively. Finally, we run Alg. 1 to maximize the probability of reaching $G^\times$ while avoiding $B^\times$, using confidence parameter $\delta/2$, state set $S^\times$, action set $A^\times$, and initial state $s_{\text{init}}^\times$. The following theorem states that Alg. 4 is regret-free.

**Theorem 5.1.** *With probability at least $(1 - \delta)$, Alg. 4 has regret $R(K) = \mathcal{O}(\sqrt{K})$.*

## 6. Discussion and Conclusions

In this paper, we proposed a regret-free algorithm for the control policy synthesis problem over unknown MDPs against infinite-horizon LTL specifications. The defined regret quantifies the accumulated deviation from the optimal probability of satisfying the LTL specification. We first propose a regret-free algorithm for until (reach-avoid) formulas, and then extend this approach to solve the synthesis problem for general LTL specifications by leveraging the knowledge of a minimum transition probability. This probability is used to design an efficient graph-learning algorithm with arbitrary precision.

### Acknowledgements

This research is supported by the following grants: EIC 101070802, ERC 101089047, DFG project 389792660 and

**Algorithm 4** Regret-free learning algorithm for general LTL specifications

---

**Input:** State and action sets $S$ and $A$, initial state $s_{\text{init}}$, LTL specification $\varphi$, confidence parameter $\delta \in (0, 1)$, minimum transition probability $p_{min} \in (0, 1)$

Construct a DRA $\mathfrak{A}_\varphi = (Q, \Sigma, \gamma, q_{\text{init}}, F)$ which accepts $\varphi$

Run Alg. 5 to get the connection graph $\chi(M)$, i.e., $\chi(M) = \mathsf{GraphLearn}(S, A, s_{\text{init}}, \delta/2, p_{min})$

Compute the graph $\chi(M_\varphi)$ using Eq. (15)

Compute MECs within $M_\varphi$ using Alg. 47 in (Baier & Katoen, 2008)

$G^\times \leftarrow \emptyset$

$B^\times \leftarrow \emptyset$

**for** MEC $C$ inside $M_\varphi$ with state space $D$ **do**

    Use Eq. (16) to check if $C$ is an AMEC

    **if** $C$ is an AMEC **then**

        $G^\times \leftarrow G^\times \cup D$

    **else**

        $B^\times \leftarrow B^\times \cup D$

    **end if**

**end for**

Run Alg. 1 with state and action sets $S^\times$ and $A^\times$, initial state $s_{\text{init}}^\times$, sets $G^\times$ and $B^\times$, confidence parameter $\delta/2$, and minimum transition probability $p_{min}$, to compute and update the policy over $M_\varphi$

---

TRR 248–CPEC.

## Impact Statement

This paper presents work whose goal is to advance the field of Machine Learning. Our work is mathematical/algorithmic. While there may be many potential societal consequences of our work, we feel none of them must be specifically highlighted here.

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

---

**Algorithm 5** Graph learning algorithm (GraphLearn)

---

**Input:** State and action sets $S$ and $A$, initial state $s_{\mathsf{init}}$, confidence $\delta \in (0, 1)$, minimum transition probability $p_{min} \in (0, 1)$
Initialization: $t \leftarrow 1$, $E \leftarrow \emptyset$, $n(s, a) = 0$ for every $(s, a) \in S \times A$, and $N(s, a, s') = 0$ for every $(s, a, s') \in S \times A \times S$
Compute a lower bound for $n^*$ using Eq. (17)
Set $L := |S| \frac{\log(\delta/6)}{\log(1 - p_{min}^{|S|})}$
**Compute policies:**
**for** $s \in S \setminus s_{\mathsf{init}}$ **do**
    Compute a policy $\pi^{(s)}$ under which $s$ is reachable from $s_{\mathsf{init}}$ with positive probability: $\pi^{(s)} = \text{Reach}(S, A, s, , \delta, p_{min})$
**end for**
**for** $(s, a) \in S \times A$ **do**
    $\pi^{(s)}(s) \leftarrow a$
    **Collect enough samples by executing the computed policies:**
    **while** $n(s, a) < n^*$ **do**
        $s_t \leftarrow s_{\mathsf{init}}$
        **for** $1 \le t \le L$ **do**
            Execute the action $a_t = \pi^{(s)}(s_t)$ and observe the next state $s_{t+1}$
            $n(s_t, a_t) \leftarrow n(s_t, a_t) + 1$
            $N(s_t, a_t, s_{t+1}) \leftarrow N(s_t, a_t, s_{t+1}) + 1$
            $t \leftarrow t + 1$
            **if** $s_t = s$ **then**
                break
            **end if**
        **end for**
    **end while**
    **Verify existence of transitions:**
    **for** $s' \in S$ **do**
        **if** $N(s, a, s') > 0$ **then**
            $E \leftarrow E \cup \{(s, a, s')\}$
        **end if**
    **end for**
**end for**
**Results:** $\chi(M) = \text{GraphLearn}(S, A, s_{\mathsf{init}}, \delta, p_{min})$

---

## A. Graph Identification

In this section, we show how to leverage the knowledge of the minimum transition probability $p_{min} \in (0, 1)$ of a given MDP $M$ to identify its underlying graph with a desired level of confidence. The following lemma provides a bound on the minimum number of samples required to verify the existence of a specific transition in $M$.

**Lemma A.1.** *(Voloshin et al., 2022) For any transition $(s, a, s') \in S \times A \times S$, let $\hat{F}_n(s'|s, a)$ denote the empirical estimation of transition probability–associated with the actual transition probability $T(s'|s, a)$–after observing $(s, a)$ for $n$ times. Given a positive lower bound over the minimum transition probability $p_{min} \in (0, 1)$ and a confidence parameter $\delta \in (0, 1)$, we have $(s, a, s') \notin E$ with confidence at least $(1 - \delta/2)$ if $\hat{F}_{n^*}(s'|s, a) = 0$, for*

$$n^* \ge \psi^{-1}(p_{min}), \tag{17}$$

*where $\psi(n) = \sqrt{\frac{1}{2}\zeta(n)} + \frac{7}{3}\zeta(n)$, and $\zeta(n) = \frac{1}{n-1} \log\left(\frac{8}{\delta} n^2 |S|^2 |A| p_{min}\right)$ if $n > 1$.*

Alg. 5 outlines our proposed algorithm to learn the graph for a given MDP. We leverage $p_{min}$ and $\delta$ to compute the minimum number of required samples $n^*$ using Eq. (17), and a horizon $L \in \mathbb{N}$, after which the execution of episodes is stopped. The process consists of two main steps: (1) for every state $s \in S$, we apply Alg. 6, that is inspired by the $\omega-$PAC algorithm (Perez et al., 2024), to obtain a policy $\pi^{(s)}$, under which $s$ is reachable from $s_{\mathsf{init}}$ with positive probability; (2) for each action $a \in A$, we execute $\pi^{(s)}$ until the pair $(s, a)$ has been visited at least $n^*$ many times. Upon reaching $(s, a)$, we collect

---

**Algorithm 6** Reachability policy synthesis for graph learning (Reach)

---

**Inputs:** State and action sets $S$ and $A$, target state $s^* \in S \setminus \{s_{\text{init}}\}$, confidence parameter $\delta \in (0, 1)$, minimum transition probability $p_{min} \in (0, 1)$

$n(s, a) \leftarrow 0$ for every $(s, a) \in S \times A$

$N(s, a, s') \leftarrow 0$ for every $(s, a, s') \in S \times A \times S$

Compute $C_3$ and $H$ using Eq. (20)

$G \leftarrow \{s^*\}$

**while** True **do**

    **for** $(s, a) \in S \times A$ **do**

        **if** $n(s, a) < c$ **then**

$$\hat{T}(s, a, s') = \begin{cases} 1 & \text{if } s' = s \\ 0 & \text{Otherwise} \end{cases}$$

          $G \leftarrow G \cup \{s\}$

        **else**

          $\hat{T}(s, a, s') = \frac{N'(s,a,s')}{n'(s,a)}$

        **end if**

    **end for**

    $\hat{M} = (S, A, \hat{T}, s_{\text{init}}, \hat{L}, \hat{AP})$, with $\hat{AP} = \{\text{Goal}\}$, and $\hat{L}(s^*) = \{\text{Goal}\}$ and $\hat{L}(s) = \emptyset$ if $s \neq s^*$

    Compute optimal policy $\pi^{(s^*)}$ for $\hat{M}$ by solving the corresponding linear program (see Sec. 10.6.1 in (Baier & Katoen, 2008))

    For $\hat{M}$, compute $S_H \subseteq S$ as the set of states that are reachable from $s_{\text{init}}$ within $2H$ time steps

    **if** $n(s, a) \geq C_3$ for every $(s, a) \in S_H \times A$ **then**

        break

    **end if**

    $s_1 \leftarrow s_{\text{init}}$

    **for** $1 \leq t \leq 2H$ **do**

        Execute the action $a_t = \pi^{(s^*)}(s_t)$ and observe the next state $s_{t+1}$

        $n(s_t, a_t) \leftarrow n(s_t, a_t) + 1$

        $N(s_t, a_t, s_{t+1}) \leftarrow N(s_t, a_t, s_{t+1}) + 1$

    **end for**

**end while**

**Results:** $\pi^{(s^*)} = \text{Reach}(S, A, s^*, \delta, p_{min})$

---

the resulting outgoing transitions by simulating the MDP. Since the initial state of the MDP is fixed at $s_{\text{init}} \neq s$, we use $\pi^{(s)}$ to reach $s$, enabling us to observe the outgoing transitions from $(s, a)$. Additionally, the horizon $L$ is used to prevent unlimited exploration if the sample trajectory is trapped in one of the MECs not containing $s$. Once $n(s, a) \geq n^*$, we verify whether the transition $(s, a, s')$ belongs to $E$, by checking if $N(s, a, s') > 0$ for every $s' \in S$.

**Sample complexity analysis.** To provide a sample complexity for Alg. 5, we need to (1) provide a bound over the minimum number of samples required by the method used for finding $\pi^{(s)}$ to enable providing a bound over the minimum number of required samples, and (2) calculate the horizon $L$ such that it offers a formal bound on the length of trajectories before getting stuck in an MEC. For (1), we must use undiscounted RL formulations such as $E^3$ (Kearns & Singh, 2002) and $\omega$-PAC (Perez et al., 2024), which explicitly handle the exploration-exploitation trade-off. For (2), given a certain confidence level, we can derive an upper bound on the number of steps after which, with high confidence, one of the MECs is reached and every state within that particular MEC is explored. The following theorem quantifies the sample complexity of Alg. 5.

**Theorem A.2.** *Let the horizon $L$ for an MDP $M$ be defined as the smallest time $t \in \mathbb{N}$ such that with probability at least $(1 - \delta/6)$ a trajectory of length $t$ starting from the initial state $s_{\text{init}}$ visits every state in some MEC within $M$. Also, let $C$ denote the minimum number of time steps after which the graph learned by Alg. 5 is correct with confidence at least $(1 - \delta)$. We have*

$$L \leq |S| \frac{\log(\delta/6)}{\log(1 - p_{min}^{|S|})} \ and \tag{18}$$

$$C \leq |S|C_1 + |S||A|LC_2, \tag{19}$$

*where*

$$C_1 := H \left\lceil \max \left( \frac{18C_3|S||A|}{p_{min}^{|S|}}, \frac{162(C_3|S||A| - \log(\delta/12))}{p_{min}^{2|S|}} \right) \right\rceil \ and$$

$$C_2 := \min \left\{ t \in \mathbb{N} \Big| \binom{t}{n^*} \left( \frac{1}{2} p_{min}^{|S|} \right)^{n^*} \left( 1 - \frac{p_{min}^{|S|}}{2} \right)^{t - n^*} > 1 - \frac{\delta}{6} \right\},$$

*with*

$$C_3 := \left\lceil \frac{-9\log(\delta/12|S||A|)}{p_{min}^{|S|}} \right\rceil, \ and \ H := |S| \frac{\log(p_{min}^{|S|}/18)}{\log(1 - p_{min}^{|S|})}. \tag{20}$$

*Proof.* Using the same reasoning as in the proof of Lem. B.1, we have $L \leq |S| \frac{\log(\delta/6)}{\log(1 - p_{min}^{|S|})}$, where the confidence level is set to $(1 - \delta/6)$. Regarding the sample complexity for computing policies $\pi^{(s)}$ for every $s \in S$, we note that either $G$ is not reachable, or the minimum reachability probability is $p_{min}^{|S|}$. Thus, to ensure that $s$ is reachable with positive probability, the computed policy must be $\varepsilon$-optimal with $\varepsilon < p_{min}^{|S|}$. By setting the confidence to $(1 - \delta/6)$ and the precision to $\varepsilon = \frac{p_{min}^{|S|}}{18}$, we can directly apply the sample complexity result from $\omega$-PAC (Perez et al., 2024) to obtain $C_1$ (see Thm. 2 in (Perez et al., 2024)). This ensures that the resulting policy achieves a positive reachability probability (greater than $p_{min}^{|S|}/2$) after $C_1$ time steps. Since we need to compute reachability policies for every state $s \in S$, this takes $|S|C_1$ time steps.

Next, $C_2$ gives the minimum number of trials after which the probability of visiting a specific state $s$ at least $n^*$ times is at least $(1 - \delta/6)$. For every state-action pair $(s, a) \in |S| \times |A|$, we must run episodes of length $L$, which requires $|S||A|LC_2$ time steps.

Since the confidence for computing $L$, $C_1$, and $C_2$ is $(1 - \delta/6)$, and for $n^*$ it is $(1 - \delta/2)$, the overall confidence becomes $(1 - \delta)$. Therefore, the total number of samples required to achieve an overall confidence of $(1 - \delta)$ is $|S|C_1 + |S||A|LC_2$. $\quad\square$

## B. Proofs

### B.1. Proof of Lem. 4.1

We adopt notation similar to that used in the proof of Lem. 3 in (Tarbouriech et al., 2020) to improve clarity. We want to bound the probability of the event $\mathcal{E}^C := \bigcup_{k=1}^{\infty} \{M \notin \mathcal{M}_k\}$. Let $B_n(s, a)$ be the $\ell_1$-ball centered at $\hat{T}_k(.|s, a)$ with radius $\beta_k(s, a)$, where $(s, a)$ is visited at least $n$ times before episode $k$. Then, $\mathcal{E}^C \subseteq \bigcup_{(s,a)} \bigcup_{n=0}^{\infty} \{T(. \mid s, a) \notin B_n(s, a)\}$. Using Boole's inequality, we get $\mathbb{P}(\mathcal{E}^C) \leq \sum_{(s,a)} \sum_{n=0}^{\infty} \mathbb{P}(T(.|s, a) \notin B_n(s, a))$.

For $n \geq 0$, define $\epsilon_n(s, a) := \sqrt{\frac{2}{n^+} \log(5|S||A|(n^+)^2(2^{(|S|+1)} - 2)/\delta)}$, where $n^+ := \max(n, 1)$. Following (Tarbouriech et al., 2020) and Weissman's inequality, we have $\epsilon_n(s, a) \leq \beta_n(s, a)$ almost surely, and for $n \geq 1$: $\mathbb{P}(T(.|s, a) \notin B_n(s, a)) \leq \delta/(5|S||A|n^2)$, and equals zero for $n = 0$.

Thus, $\mathbb{P}(\mathcal{E}^C) \leq \sum_{(s,a)} \sum_{n=1}^{\infty} \frac{\delta}{5n^2|S||A|} = \frac{\pi^2}{6} \cdot \frac{\delta}{5} < \delta/3$. This concludes the proof.

### B.2. Proof of Lem. 4.4

In order to prove Lem. 4.4, we need to first ensure the existence of a (high-probability) bound on the number of steps required to reach $G$ from states in $S \setminus B$, when we enable reset. Note that the set $B$ also includes every MEC whose intersection with $G$ is empty. In App. A, we discuss details of an algorithm for learning the MDP graph $\chi(M)$ up to any desired confidence, using the knowledge of the minimum transition probability $p_{min}$. Given $\chi(M)$, we can efficiently

identify all MECs within $M$ that do not intersect with $G$ and include them in $B$. The following lemma provides an upper bound for the time required to reach $G$ from any state in $S \setminus B$, when a reset transition is enabled upon visiting states in $B$.

**Lemma B.1.** *Let $\tilde{\lambda}_k(s)$ denote the time to reach $G$ from state $s$ in an (artificial) MDP $\tilde{M}'_k$, obtained by connecting $B$ to $s_{\text{init}}$ with probability 1 in $\tilde{M}_k$. Then, with confidence at least $(1 - \delta/6)$, we have for every $s \in S \setminus B$ that*

$$\tilde{\lambda}_k(s) \leq \Lambda := |S| \frac{\log(\delta/6)}{\log(1 - p_{min}^{|S|})}. \tag{21}$$

*Proof.* First, we note that by including every state from which $G$ is unreachable into $B$, it is guaranteed that policy $\tilde{\pi}_k(s)$ takes every $s \in S \setminus B$ to $G$, with probability at least $p_{min}^{|S|}$. This is ensured by Eq. (6), since $v^*(s) \geq p_{min}^{|S|}$ for every $s \in S \setminus B$. Now, enabling the reset transition from $B$ to $s_{\text{init}}$ with probability 1, under $\tilde{\pi}_k$, $G$ becomes reachable from every state $s \in S$ with probability 1. In the worst-case scenario, every state in the corresponding Markov chain must be visited at least once. Visiting every state requires following a path of length at most $|S|$, which occurs with probability $p_{min}^{|S|}$. After $l$ attempts of traversing this path, the probability of success at least once is given by $1 - (1 - p_{min}^{|S|})^l$. If $l \geq \frac{\log(\delta/6)}{\log(1 - p_{min}^{|S|})}$, then $1 - (1 - p_{min}^{|S|})^l \geq 1 - \delta/6$. Finally, since each of the $l$ attempts takes $|S|$ steps in the worst case, the total number of steps is bounded by $|S| \times l = |S| \frac{\log(\delta/6)}{\log(1 - p_{min}^{|S|})}$. Furthermore, since $\tilde{M}_k$ is chosen optimistically, the time to reach $G$ in $\tilde{M}_k$ is no greater than that in the true MDP $M$. Therefore, with probability at least $(1 - \delta/6)$, we have that $\tilde{\lambda}_k(s) \leq \Lambda$.

$\square$

Now, we are ready to prove Lem. 4.4. First, using the Markov's inequality (since $x \mapsto x^r$ is a non-decreasing mapping for non-negative reals), we have

$$\mathbb{P}(\tilde{\lambda}_k(s) \geq H_k - 1) \leq \frac{\mathbb{E}[\tilde{\lambda}_k(s)^r]}{(H_k - 1)^r}.$$

Now, we note that by Lem. 15 in (Tarbouriech et al., 2020), we have if $\tilde{\lambda}_k(s) \leq \lambda$ for every $s \in S \setminus (G \cup B)$ and $\lambda \geq 2$, then

$$\mathbb{E}(\tilde{\lambda}_k(s)^r) \leq 2(r\lambda)^r,$$

for any $r \geq 1$. Therefore, substituting $\lambda$ with $\Lambda$ (defined in Eq. (21)), we will have

$$\mathbb{P}(\tilde{\lambda}_k(s) \geq H_k - 1) \leq \frac{2(r\Lambda)^r}{(H_k - 1)^r}, \tag{22}$$

Note that there exists $y \in S$ such that

$$\|\tilde{Q}^{H_k - 2}\|_\infty = \mathbf{1}_y^\top \tilde{Q}^{H_k - 2} \mathbf{1} = \mathbb{P}(\tilde{\lambda}_k(y) > H_k - 2) = \mathbb{P}(\tilde{\lambda}_k(y) \geq H_k - 2). \tag{23}$$

By definition of $H_k$ we have $\|\tilde{Q}^{H_k - 2}\|_\infty > 1/\sqrt{k}$. Combining this with Eqs. (22), (23) yields

$$\frac{2(r\Lambda)^r}{(H_k - 1)^r} > 1/\sqrt{k},$$

which implies that

$$H_k - 1 < r\Lambda(2\sqrt{k})^{1/r}.$$

By selecting $r = \left\lceil \log(2\sqrt{k}) \right\rceil$, we get

$$H_k - 1 < \lceil \log(2\sqrt{k}) \rceil \Lambda (2\sqrt{k})^{\frac{1}{\lceil \log(2\sqrt{k}) \rceil}} \leq \left\lceil 3\Lambda \log(2\sqrt{k}) \right\rceil.$$

Hence

$$\alpha_K \leq \left\lceil 3\Lambda \log(2\sqrt{K}) \right\rceil.$$

Since $\tilde{\lambda}_k(s) \leq \Lambda$ holds with confidence at least $(1 - \delta/6)$, the overall confidence is also maintained at a level at least $(1 - \delta/6)$.

## B.3. Proof of Lem. 4.5

In order to prove Lem. 4.5, we make use of the Azuma-Hoeffding inequality, which is stated below for the sake of completeness.

**Lemma B.2** (Azuma-Hoeffding inequality, Hoeffding 1963)**.** *Let $X_1, X_2, \ldots$ be a martingale difference sequence with $|X_l| \leq c$ for all $l$. Then for all $\gamma > 0$ and $n \in N$,*

$$\mathbb{P}\{\sum_{l=1}^{n} X_l \geq \gamma\} \leq \exp(-\frac{\gamma^2}{2nc^2}).$$

Now, we are ready to prove Lem. 4.5. We proceed by showing why $\sum_{k=1}^{K} \Delta_k^{\text{f1}}$ grows sublinearly with $K$. In order to reformulate the regret, we define the following reward function $r\colon S \to \{0, 1\}$

$$r(s) = \begin{cases} 0 & s \notin G \\ 1 & s \in G. \end{cases} \tag{24}$$

Further, for the time step $h$ within episode $k$ we define

$$\Theta_{k,h}(s_{k,h}) := \tilde{v}_k(s_{k,h}) - \sum_{t=h}^{H_{k,\mathcal{I}_k(h)}-1} r(s_{k,t})$$

where $\mathcal{I}_k\colon [1; H_k] \to [1; I_k]$ maps the time points in episode $k$ into the corresponding interval, and $H_{k,i}$ denotes the length of the $i^{th}$ interval within the $k^{th}$ episode for $1 \leq i \leq I_k$ and $H_{k,0} = 1$. Therefore, we have

$$\sum_{k=1}^{K} \Delta_k^{\text{f1}} = \sum_{k=1}^{K} \sum_{i=1}^{I_k} \Theta_{k,H_{k,i-1}}(s_{k,H_{k,i-1}}).$$

Let us further define

$$\Phi_{k,h} := \tilde{v}_k(s_{k,h+1}) - \sum_{y \in S} p(y \mid s_{k,h}, \tilde{\pi}_k(s_{k,h}))\tilde{v}_k(y), \tag{25}$$

where $p(.|.,.)$ corresponds to the transition probability in the true MDP $M$. Similarly, we denote by $\tilde{p}_k(.|.,.)$ for the transition probability in the optimistic MDP $\tilde{M}_k$. Note that for $s_{k,h} \in G \cup B$, we have $\Theta_{k,h}(s_{k,h}) = 0$. For $s_{k,h} \notin G \cup B$, we have

$$\Theta_{k,h}(s_{k,h}) = \tilde{v}_k(s_{k,h}) - \sum_{t=h}^{H_{k,\mathcal{I}_k(h)}-1} r(s_{k,t}) \leq \tilde{\mathcal{L}}_k \tilde{v}_k(s_{k,h}) + \varepsilon_k - \sum_{t=h}^{H_{k,\mathcal{I}_k(h)}-1} r(s_{k,t})$$

$$= \sum_{y \in S} \tilde{p}_k(y \mid s_{k,h}, \tilde{\pi}_k(s_{k,h}))\tilde{v}_k(y) + \varepsilon_k - r(s_{k,h}) - \sum_{t=h+1}^{H_{k,\mathcal{I}_k(h)}-1} r(s_{k,t})$$

$$= \sum_{y \in S} (\tilde{p}_k(y \mid s_{k,h}, \tilde{\pi}_k(s_{k,h})) - p(y \mid s_{k,h}, \tilde{\pi}_k(s_{k,h})))\tilde{v}_k(y)$$

$$+ \sum_{y \in S} p(y \mid s_{k,h}, \tilde{\pi}_k(s_{k,h}))\tilde{v}_k(y) + \varepsilon_k - \sum_{t=h+1}^{H_{k,\mathcal{I}_k(h)}-1} r(s_{k,t})$$

$$\leq 2\beta_k(s_{k,h}, \tilde{\pi}_k(s_{k,h})) \times 1 + (\sum_{y \in S} p(y \mid s_{k,h}, \tilde{\pi}_k(s_{k,h}))\tilde{v}_k(y) - \tilde{v}_k(s_{k,h+1}))$$

$$+ \varepsilon_k + (\tilde{v}_k(s_{k,h+1}) - \sum_{t=h+1}^{H_{k,\mathcal{I}_k(h)}-1} r(s_{k,t}))$$

$$\leq 2\beta_k(s_{k,h}, \tilde{\pi}_k(s_{k,h})) + \Phi_{k,h} + \varepsilon_k + \Theta_{k,h+1}(s_{k,h+1}),$$

where the first inequality follows from the termination condition of Alg. 2, the fact that $r(s_{k,t}) = 0$ for every $s_{k,t} \notin G \cup B$ for the second equality, $\tilde{v}_k(y) \leq 1$ for every $y \in S$ for the third equality, definition of $\beta_k$ (Eq. (4)) for the second inequality, and definition of $\Phi_{k,h}$ (Eq. (25)) for the last inequality. Also, note that By telescopic sum we get

$$\Theta_{k,H_{k,i}}(s_{k,H_{k,i}}) = \sum_{h=H_{k,i}}^{H_{k,i+1}-2} (\Theta_{k,h}(s_{k,h}) - \Theta_{k,h+1}(s_{k,h+1})) + \Theta_{k,H_{k,i+1}-1}(s_{k,H_{k,i+1}-1})$$

$$\leq \sum_{h=1}^{H_{k,i+1}-2} (2\beta_k(s_{k,h}, \tilde{\pi}_k(s_{k,h})) + \Phi_{k,h} + \varepsilon_k) + \Theta_{k,H_{k,i+1}-1}(s_{k,H_{k,i+1}-1})$$

$$\leq \sum_{h=1}^{H_{k,i+1}-2} 2\beta_k(s_{k,h}, \tilde{\pi}_k(s_{k,h})) + \sum_{h=1}^{H_{k,i+1}-2} \Phi_{k,h} + H_{k,i+1}\varepsilon_k,$$

where we used the arguments we made in the previous step of the proof to establish the first inequality, the fact that $\Theta_{k,H_{k,i+1}-1}(s_{k,H_{k,i+1}-1}) = 0$ by definition as $s_{k,t_{k,i+1}-1} \in G \cup B$ for every $1 \leq i \leq I_k$ for the last inequality. By summing over all of the episodes we have

$$\sum_{k=1}^{K} \sum_{i=1}^{I_k} \Theta_{k,H_{k,i-1}}(s_{k,H_{k,i-1}})$$

$$\leq \sum_{k=1}^{K} \sum_{i=1}^{I_k} \sum_{h=H_{k,i-1}}^{H_{k,i}-1} \Phi_{k,h} + 2 \sum_{k=1}^{K} \sum_{i=1}^{I_k} \sum_{h=H_{k,i-1}}^{H_{k,i}-1} \beta_k(s_{k,h}, \tilde{\pi}_k(s_{k,h})) + \sum_{k=1}^{K} H_k \varepsilon_k. \tag{26}$$

In order to bound the first term, we note that

$$\sum_{k=1}^{K} \sum_{i=1}^{I_k} \sum_{h=H_{k,i-1}}^{H_{k,i}-1} \Phi_{k,h} = \sum_{k=1}^{K} \sum_{h=t_k}^{t_{k+1}-1} \Phi_{k,h}.$$

Therefore, we can write

$$\mathbb{P}\left( \sum_{k=1}^{K} \sum_{h=t_k}^{t_{k+1}-1} \Phi_{k,h} \geq 2\sqrt{2(\sum_{k=1}^{K} H_k) \log\left( \frac{2(\sum_{k=1}^{K} H_k)^2}{\delta} \right)} \right)$$

$$\leq \mathbb{P}\left( \sum_{k=1}^{K} \sum_{h=t_k}^{t_{k+1}-1} \Phi_{k,h} \geq 2\sqrt{2n \log\left( \frac{2n^2}{\delta} \right)} \cap \sum_{k=1}^{K} H_k = n \right).$$

Let $\mathcal{G}_q$ denote the history of all random events up to (and including) step $h$ of episode $k$, i.e., $q = \sum_{k'=1}^{k-1} H_{k'} + h$. We have $\mathbb{E}(\Phi_{k,h} \mid \mathcal{G}_q) = 0$, and furthermore $H_k$ is selected at the beginning of episode $k$, and so it is adapted with respect to $\mathcal{G}_q$. Hence $\Phi_{k,h}$ is a martingale difference with $|\Phi_{k,h}| \leq 1$. Therefore, by Azuma-Hoeffding's inequality, we have with probability $1 - \frac{\delta}{3}$

$$\sum_{k=1}^{K} \sum_{h=t_k}^{t_{k+1}-1} \Phi_{k,h} \leq 2\sqrt{2\left( \sum_{k=1}^{K} H_k \right) \log\left( \frac{6(\sum_{k=1}^{K} H_k)^2}{\delta} \right)}.$$

Now, we proceed to bound the second term in Eq. (26). We can write

$$\sum_{k=1}^{K} \sum_{h=1}^{H_k-1} \frac{1}{\sqrt{N_k^+(s_{k,h}, \tilde{\pi}_k(s_{k,h}))}} \leq \sum_{s,a} \sum_{n=1}^{N_K^+} \sqrt{\frac{1}{n}} \leq 2\sqrt{|S||A|} \sqrt{\sum_{s,a} N_K^+(s,a)} \leq 2\sqrt{|S||A|t_K}.$$

Therefore, we obtain

$$\sum_{k=1}^{K} \sum_{h=1}^{H_k-1} \beta_k(s_{k,h}, \tilde{\pi}_k(s_{k,h})) \leq 2|S|\sqrt{8|A|t_K \log\left( \frac{2|A|t_K}{\delta} \right)}.$$

Finally, we bound the last term in Eq. (26).

$$\sum_{k=1}^{K} H_k \varepsilon_k \leq \sum_{t=1}^{T_{K,1}} \frac{\alpha_K}{2t} \leq \frac{\alpha_K}{2}(1 + \log(K\alpha_K)),$$

where $\alpha_K = \max_{1 \leq k \leq K} H_k$. Putting everything together yields that inequality (12) holds with probability at least $1 - 5\delta/6$.

## B.4. Proof of Lem. 4.6

We define $X_{k,i} := v_k(s_{\text{init}}) - v_{k,i}(s_{\text{init}})$. Note that $\mathbb{E}(X_{k,i}) = 0$ and $|X_{k,i}| \leq 1$ for every $1 \leq k \leq K$ and $1 \leq i \leq I_k$. We have

$$\Delta_k^{\text{f2}} = \sum_{i=1}^{I_k} v_k(s_{\text{init}}) - v_{k,i}(s_{\text{init}}) = \sum_{i=1}^{I_k} X_{k,i}.$$

Therefore by application of the Azuma-Hoeffding lemma and using the fact that $K\alpha_k \geq \sum_{k=1}^{K} I_k$ we get

$$\mathbb{P}\left[\sum_{k=1}^{K} \Delta_k^{\text{f2}} \geq \sqrt{2K\alpha_K \log(\frac{6}{\delta})}\right] \leq \exp\left[-\frac{2K\alpha_K \log(\frac{6}{\delta})}{2\sum_{k=1}^{K} I_k}\right] \leq \exp\left[-\log(\frac{6}{\delta})\right] = \delta/6.$$

In Lem. 4.4, we have already shown that $\alpha_K$ grows logarithmically with $K$ which proves that $\sum_{k=1}^{K} \Delta_k^{\text{f2}}$ grows sublinearly with $K$ with confidence at least $(1 - \delta/6)$.

## B.5. Proof of Lem. 4.7

Let $F_K := \sum_{k=1}^{K} \Delta_k^{\text{s}}$ and $\lambda_k$ and $\tilde{\lambda}_k$ denote the hitting times of policy $\tilde{\pi}_k$ in the true and optimistic models, respectively. We define

$$\Gamma_{k,h}(s_{k,h}) = \mathbf{1}_{\lambda_k(s_{k,h}) > H_k - h} - \mathbb{P}(\tilde{\lambda}_k(s_{k,h}) > H_k - h).$$

Note that we have

$$F_K = \sum_{k=1}^{K} \mathbf{1}_{\lambda_k(s_{k,1}) > H_k - 1} = \sum_{k=1}^{K} \Gamma_{k,1}(s_{k,1}) + \sum_{k=1}^{K} \mathbb{P}(\tilde{\lambda}_k(s_{\text{init}}) > H_k - 1).$$

Let $\tilde{p}'_k(.|.,.)$ denote the transition probability in the optimistic model $\tilde{M}'_k$ (that is the optimistic MDP constructed from $\tilde{M}_k$ by connecting states in $B$ into $s_{\text{init}}$). Similarly, let $p'(.|.,.)$ denote the transition probability in the MDP $M'_\varphi$ (that is the MDP constructed from $M_\varphi$ by connecting states in $B$ into $s_{\text{init}}$). Since for $1 \leq h \leq H_k - 1$, $\mathbf{1}_{\lambda_k(s_{k,h}) > H_k - h} = \mathbf{1}_{\lambda_k(s_{k,h+1}) > H_k - h - 1}$ we have

$$\Gamma_{k,h}(s_{k,h}) = \mathbf{1}_{\lambda_k(s_{k,h+1}) > H_k - h - 1} - \sum_{y \in S} \tilde{p}'_k(y \mid s_{k,h}, \tilde{\pi}_k(s_{k,h}))\mathbb{P}(\tilde{\lambda}_k(y) > H_k - h - 1)$$

$$\leq \mathbf{1}_{\lambda_k(s_{k,h+1}) > H_k - h - 1} - (\sum_{y \in S} \tilde{p}'_k(y \mid s_{k,h}, \tilde{\pi}_k(s_{k,h})) - p'(y \mid s_{k,h}, \tilde{\pi}_k(s_{k,h})))\mathbb{P}(\tilde{\lambda}_k(y) > H_k - h - 1)$$

$$- \sum_{y \in S} p'(y \mid s_{k,h}, \tilde{\pi}_k(s_{k,h}))\mathbb{P}(\tilde{\lambda}_k(y) > H_k - h - 1)$$

$$\leq \mathbf{1}_{\lambda_k(s_{k,h+1}) > H_k - h - 1} + 2\beta_k(s_{k,h}, \tilde{\pi}_k(s_{k,h})) - \sum_{y \in S} p'(y \mid s_{k,h}, \tilde{\pi}_k(s_{k,h}))\mathbb{P}(\tilde{\lambda}_k(y) > H_k - h - 1)$$

$$= \Gamma_{k,h+1}(s_{k,h+1}) + \psi_{k,h} + 2\beta_k(s_{k,h}, \tilde{\pi}_k(s_{k,h})),$$

where we established the first inequality by adding and subtracting the term $p'(y \mid s_{k,h}, \tilde{\pi}_k(s_{k,h}))\mathbb{P}(\tilde{\lambda}_k(y) > H_k - h - 1)$, the second inequality by using the definition of $\beta_k$ (Eq. (3)), and the last equality by using the following definition

$$\psi_{k,h} = \mathbb{P}(\tilde{\lambda}_k(s_{k,h+1}) > H_k - h - 1) - \sum_{y \in S} p'(y \mid s_{k,h}, \tilde{\pi}_k(s_{k,h}))\mathbb{P}(\tilde{\lambda}_k(y) > H_k - h - 1).$$

Also, we have

$$\Gamma_{k,H_k}(s_{k,H_k}) = \mathbf{1}_{\lambda_k(s_{k,H_k})>0} - \mathbb{P}(\tilde{\lambda}_k(s_{k,H_k} > 0)) = \mathbf{1}_{\lambda_k(s_{k,H_k})>0} - \mathbf{1}_{\tilde{\lambda}_k(s_{k,H_k}>0)} = \mathbf{1}_{s_{k,H_k}\notin G} - \mathbf{1}_{s_{k,H_k}\notin G} = 0.$$

Using the telescopic sum we get

$$\Gamma_{k,1}(s_{k,1}) = \sum_{h=1}^{H_k-1}\left(\Gamma_{k,h}(s_{k,h}) - \Gamma_{k,h+1}(s_{k,h+1})\right) + \Gamma_{k,H_k}(s_{k,H_k}) \leq \sum_{h=1}^{H_k-1}\psi_{k,h} + 2\sum_{h=1}^{H_k-1}\beta_k(s_{k,h},\tilde{\pi}_k(s_{k,h})),$$

where the last inequality is achieved by knowing $\Gamma_{k,H_k}(s_{k,H_k}) = 0$. Therefore, by summing over all episodes we get

$$F_K \leq \sum_{k=1}^{K}\sum_{h=1}^{H_k-1}\psi_{k,h} + 2\sum_{k=1}^{K}\sum_{h=1}^{H_k-1}\beta_k(s_{k,h},\tilde{\pi}_k(s_{k,h})) + \sum_{k=1}^{K}\mathbb{P}(\tilde{\lambda}_k(s_{\text{init}}) > H_k - 1).$$

Let $\mathcal{G}_q$ denote the history of all random events up to (and including) step $h$ of episode $k$, i.e., $q = \sum_{k'=1}^{k-1} H_k + h$. We have $\mathbb{E}(\psi_{k,h} \mid \mathcal{G}_q) = 0$, and furthermore $H_k$ is selected at the beginning of episode $k$, and so it is adapted with respect to $\mathcal{G}_q$. Hence $\psi_{k,h}$ is a martingale difference with $|\psi_{k,h}| \leq 1$. Therefore, using similar arguments as in proof of Lem. 4.5, by Azuma-Hoeffding's inequality, we have with probability $1 - \frac{2\delta}{3}$

$$\sum_{k=1}^{K}\sum_{h=1}^{H_k-1}\psi_{k,h} \leq 2\sqrt{2\left(\sum_{k=1}^{K}H_k\right)\log\left(\frac{3(\sum_{k=1}^{K}H_k)^2}{\delta}\right)} \leq 2\sqrt{2K\alpha_K\log\left(\frac{3(K\alpha_K)^2}{\delta}\right)}.$$

Further, in the same vein as proof of Lem. 4.5 we have

$$\sum_{k=1}^{K}\sum_{h=1}^{H_k-1}\beta_k(s_{k,h},\tilde{\pi}_k(s_{k,h})) \leq 2|S|\sqrt{8|A|K\alpha_K\log\left(\frac{2|A|K\alpha_K}{\delta}\right)}.$$

Now, we need to bound $\sum_{k=1}^{K}\mathbb{P}(\tilde{\lambda}_k(s_{\text{init}}) > H_k - 1)$. Using Thm. 2.5.3 in (Latouche & Ramaswami, 1999), we have

$$\sum_{k=1}^{K}\mathbb{P}(\tilde{\lambda}_k(s_{\text{init}}) > H_k - 1) = \sum_{k=1}^{K}\mathbf{1}_{s_{\text{init}}}\tilde{Q}_k^{H_k-1}\mathbf{1},$$

where $\mathbf{1}_s$ denotes the $|S| - 1$-sized one-hot vector at the position of state $s \in S$. Finally, from Holder's inequality, we have

$$\sum_{k=1}^{K}\mathbb{P}(\tilde{\lambda}_k(s_{\text{init}}) > H_k - 1) = \sum_{k=1}^{K}\mathbf{1}_{s_{\text{init}}}\tilde{Q}_k^{H_k-1}\mathbf{1}$$

$$\leq \sum_{k=1}^{K}\|\mathbf{1}_{s_{\text{init}}}\|_1\|\tilde{Q}_k^{H_k-1}\mathbf{1}\|_\infty \leq \sum_{k=1}^{K}\|\tilde{Q}_k^{H_k-1}\mathbf{1}\|_\infty.$$

Therefore, by the choice of $H_k = \min\{n > 1 \mid \|\tilde{Q}_k^n\|_\infty \leq \frac{1}{\sqrt{k}}\}$ we get

$$\sum_{k=1}^{K}\mathbb{P}(\tilde{\lambda}_k(s_{\text{init}}) > H_k - 1) \leq \sum_{k=1}^{K}\frac{1}{\sqrt{k}} \leq 2\sqrt{K}.$$

Putting everything together yields that inequality (14) holds with probability at least $1 - \delta$.

### B.6. Proof of Thm. 4.2

Let $\mathcal{Y}^{(f,1)}$, $\mathcal{Y}^{(f,2)}$, and $\mathcal{Y}^{(s)}$ denote the events under which Eqs. (12), (13), and (14) hold, respectively. By Lems. 4.5 and 4.6, we have $\mathbb{P}(\mathcal{Y}^{(f,1)} \cap \mathcal{Y}^{(f,2)}) \geq 1 - 5\delta/6 - \delta/6 = 1 - \delta$. Moreover, from Lem. 4.7, we know that $\mathbb{P}(\mathcal{Y}^{(s)}) \geq 1 - \delta$.

Now, let $\mathcal{Y}_r$ denote the event under which inequality (9) holds. Since $\mathcal{Y}^{(f,1)} \cap \mathcal{Y}^{(f,2)} \cap \mathcal{Y}^{(s)} \subseteq \mathcal{Y}_r$, it follows that $\mathbb{P}(\mathcal{Y}_r) \geq 1 - \delta - \delta = 1 - 2\delta$.

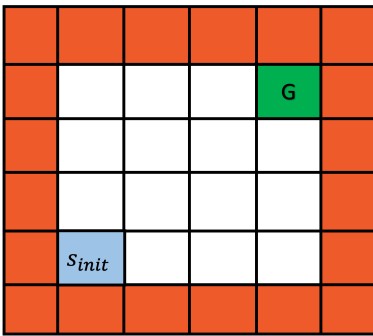

Figure 2: Map of the gridworld example with $l = 6$. The blue and green cells define the initial state ($s_{\text{init}}$) and target state ($G$), respectively. The red cells correspond to walls. The objective of the agent is to reach $G$ without hitting the walls.

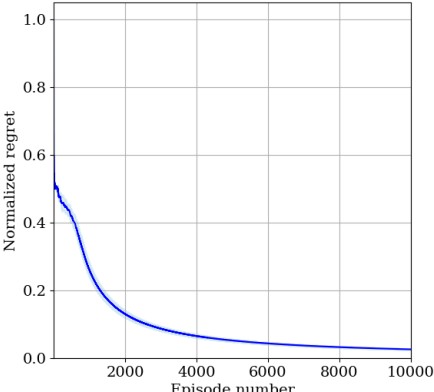

Figure 3: Variations of the empirical normalized regret, $R(K)/K$, when our proposed algorithm is applied to the gridworld example with $l = 6$.

### B.7. Proof of Thm. 5.1

Let $\mathcal{Y}_g$ and $\mathcal{Y}_r$ denote the events, under which the learned graph is correct, and the bound over the regret given by Eq. (9) holds (after substituting $S, A, \Lambda$ with $S^\times, A^\times, \Lambda^\times$, respectively). From the results of Thms. 4.2 and A.2, we have $\mathbb{P}(\mathcal{Y}_g) = \mathbb{P}(\mathcal{Y}_r) \geq 1 - \delta/2$. Then,

$$\mathbb{P}(\mathcal{Y}_g \cap \mathcal{Y}_r) = 1 - \mathbb{P}(\mathcal{Y}_g \cup \mathcal{Y}_r) \geq 1 - \delta/2 - \delta/2 = 1 - \delta.$$

## C. Experimental Evaluation

In this section, we evaluate an implementation of our algorithm. The experiments are performed on a laptop with core i7 CPU at 3.10GHz, with 8GB of RAM. We implement Alg. 1 to obtain the results, assuming that the underlying graph structure is known. This assumption is justified by the fact that, for a fixed system dynamics, Alg. 5 needs to be executed only once to learn the corresponding graph with the desired confidence. The resulting graph can then be reused for verifying any LTL specification.

We considered a reach-avoid policy synthesis problem in the gridworld example described in Fig. 2. The world is characterized by $l \in \mathbb{N}_{\geq 4}$ that denotes the number of cells per column and row. The agent can move using the cardinal directions, i.e., $A = \{\, right, left, up, down \,\}$. Movement along an intended direction succeeds with probability 0.9 and fails

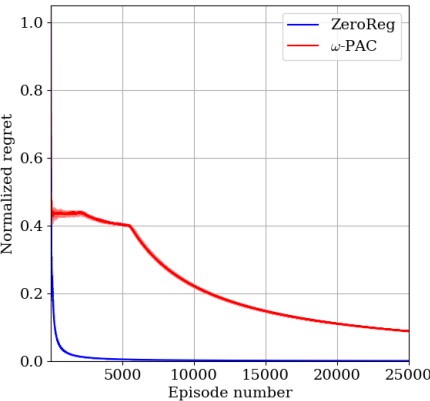

Figure 4: Comparison of the empirical normalized regret between our proposed regret-free algorithm and the $\omega$-PAC algorithm (Perez et al., 2024) for the gridworld example with $l = 4$.

with probability $0.1$. In case of failure, the agent does not move. Walls are considered to be absorbing, i.e., the agent will not be able to move after hitting a wall. We have conducted experiments to (1) evaluate the empirical performance of our algorithmcm, (2) observe how episode length vary throughout the run of our algorithm, and (3) assess the sample complexity of our method.

**Empirical performance:** Fig. 3 illustrates the variations of empirical mean for the normalized regret $R(K)/K$ for our regret-free algorithm which is run for the gridworld example with $l = 6$. We set $\delta = 0.1$ over 10 runs. Furthermore, we group all of the cells associated with the wall into an absorbing state $B$, such that we have $|S| = 17$ and $|A| = 4$. The considered specification is $\varphi = \neg\mathsf{Avoid} \ U \ \mathsf{Goal}$, where $G$ is marked as $\mathsf{Goal}$ and walls are marked as $\mathsf{Avoid}$. It can be observed that the empirical mean of the regret drops very quickly, which implies that the algorithm successfully finds an optimal policy within the few first episodes.

We also compare the performance of our proposed method with the $\omega$-PAC algorithm of (Perez et al., 2024), which is the only existing method that supports guaranteed policy synthesis against infinite-horizon temporal specifications. The $\omega$-PAC algorithm takes a confidence parameter $\delta \in (0, 1)$ and a precision parameter $\varepsilon \in (0, 1)$ and provides a policy which has $\varepsilon$-optimal satisfaction probability with confidence $(1 - \delta)$. Fig. 4 illustrates that our proposed algorithm converges much faster. We believe that this is because our algorithm uses the intermediate confidence bounds, while the $\omega$-PAC algorithm waits until enough many samples are collected, and only then updates its policy.

**Episode length variations:** Fig. 5 illustrates the variations in $H_k$ for different episodes. Initially, our algorithm assigns very small values to $H_k$, since the expected time to reach $G$ in the optimistic MDP is small. As the empirical transition probabilities become more precise, the estimation over the expected time to reach $G$ takes more accurate values.

**Sample complexity:** Although our method and $\omega$-PAC algorithm provide different guarantees, we relate them through definition of a related complexity metric. The sample complexity of the $\omega$-PAC algorithm is characterized with $C$ that is the number of learning episodes with non-$\varepsilon$-optimal satisfaction probability. We define $k^*_{reg}(\delta, \varepsilon)$ as the smallest number of episodes $k$ for which our regret-free algorithm satisfies $\frac{R(k)}{k} \leq \varepsilon$ with confidence $(1-\delta)$. Furthermore, we define $k^*_{PAC}(\delta, \varepsilon)$ as the minimum number of episodes after which the $\omega$-PAC algorithm satisfies $\frac{C_k}{k} \leq \varepsilon$ with confidence $(1 - \delta)$. Fig. 6 illustrates the variations of $k^*_{reg}(0.1, 0.1)$ and $k^*_{PAC}(0.1, 0.1)$ for gridworld example with $4 \leq l \leq 16$. Note that changes in $l$ influences the size of the state space ($|S| = (l - 2)^2 + 1$) and also the (minimum) $\varepsilon$-recurrence time $T_\varepsilon = (l - 2)^2 + 1$, which is required by the $\omega$-PAC algorithm. Furthermore, we set $p_{min} = 0.01$. It can be observed that our algorithm provides a tighter bound specially for the larger examples.

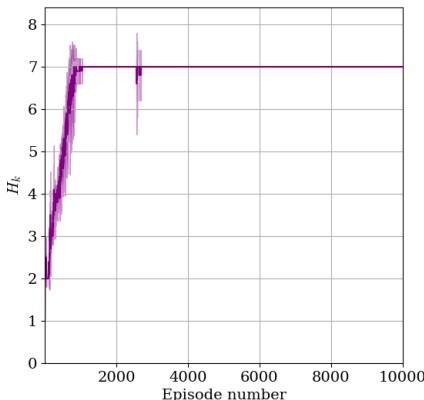

Figure 5: Variations of the computed deadline, $H_k$, when our proposed algorithm is applied to the gridworld example with $l = 6$.

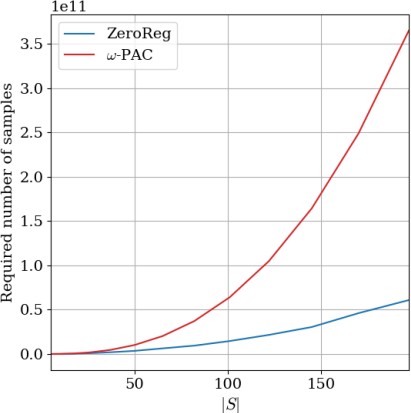

Figure 6: Comparison of the theoretical sample complexities for our proposed algorithm and the $\omega$-PAC algorithm (Perez et al., 2024) for the gridworld example with various sizes ($4 \leq l \leq 16$).

