# OpenReview forum: "Regret-Free Reinforcement Learning for Temporal Logic Specifications"
_ICML.cc/2025/Conference — ICML 2025 poster_

### Official Review · Reviewer_uDpf · 2025-03-09

**Overall Recommendation:** 3

**Summary:**

The paper tackles reinforcement learning (RL) under linear temporal logic (LTL) specifications in unknown Markov decision processes (MDPs). The primary goal is to guarantee sublinear regret with respect to the (unknown) optimal probability of satisfying an LTL property. From classic reach-avoid setting, author also extend to a more general setting using LTL property to a deterministic Rabin automaton.

**Claims And Evidence:**

Claims are all supported by solid proof and toy example experiment backup.

**Essential References Not Discussed:**

Most of the works review know has been properly cited in the paper.

**Experimental Designs Or Analyses:**

Yes, the experiment makes sense under the basic tabular setting. However, it would be great if author can conduct experiments on more complex setting, eg. more complicated map?

**Methods And Evaluation Criteria:**

Methods and evaluation criteria make sense. Since this approach is under tabular MDP setting, experiment seems fairly easy but the major contribution is on the theory side.

**Other Comments Or Suggestions:**

1. It would be better to improve the writing for section 4.1. Each sub-section in 4.1 seems quite detached from each other.

**Other Strengths And Weaknesses:**

1. For the reach-avoid case, the authors prove a regret bound on the order of $O(\sqrt{K})$ for K episodes, with high probability.
2. By extension, for the general LTL setting (using the product MDP trick), the same $O(\sqrt{K})$ scaling holds again, under certain assumptions such as known pmin and the ability to treat non-accepting states or MECs as “resets.”
3. They mention gridworld-type experiments showing that their approach converges faster than a purely PAC-based method from prior literature. While not a large-scale experiment, it demonstrates that the theoretical advantage—sublinear regret—can also manifest as faster finite-time progress in practice.
4. The biggest step forward is a proven finite-time, sublinear regret guarantee for LTL tasks in unknown MDPs, extending ideas from optimistic exploration to logic-based specifications.
5. Once they can identify the sets of accepting vs. rejecting MECs (via pmin and some exploration episodes), they unify a wide class of LTL formulas under the same approach.
6. A key assumption is that pmin (a positive lower bound on all nonzero transition probabilities) is known in advance. This is not always realistic.

**Questions For Authors:**

1. reviewer is curious in eq4 how the bound of plausible MDPs is chosen? And what is the intuition behind current MDPs' transition bound?

**Relation To Broader Scientific Literature:**

This work borrows idea from classic controller synthesis to

**Theoretical Claims:**

I have walked through proof of thm 4.4 which makes sense to me, but I'm curious of the definition of transition function in eq4.

---

> ### Author Rebuttal · Authors · 2025-03-31
>
> We thank you for the detailed and useful feedback! We plan to improve our paper taking your comments into account. Our response to your specific questions are summarized as follows.
>
> **Discussion on the practicality of known lower bounds for transition probabilities:**
>
> We agree with the reviewer that in a practice deriving $p_{min}$ might be challenging. However, all we need is to know a positive lower bound for the minimum transition probability, and not its exact value. Further, the assumption of knowing $p_{min}$ is strictly weaker than the other typical assumptions such as knowing the underlying connection graph of MDP, and there are already negative results in the literature regarding the consequences of not knowing $p_{min}$ for learning against infinite-horizon specifications. Consider a simple MDP with a state set $S = \set{s_{\text{init}}, G}$ and a singleton action set $A = \set{a}$, where the only outgoing transition from the initial state $s_{\text{init}}$ has transition probability $T(s_{\text{init}}, a, G) = \varepsilon$. It is easy to observe that, while the optimal value is one, selecting any $p_{min} > \varepsilon$ results in learning an MDP where the optimal value becomes zero. Hence, knowledge of $p_{min}$ is crucial for effectively learning in infinite-horizon tasks. In practice, domain-specific knowledge of the target system can be helpful in order to choose a sensible value for $p_{min}$.
>
>
> **Writing of section 4.1:**
>
> Section 4.1 is specifically dedicated to explaining various components of the main algorithm (Algorithm 1). In particular, the following components of Algorithm 1 are discussed in detail: (1) the application of Equation (4) for computing confidence intervals, which are then used to construct interval MDPs from the observed data; (2) the operation of our proposed Extended Value Iteration method (Algorithm 2); and (3) the use of Equation (9) to compute episode-specific deadlines. These explanations are provided in the respective subsections of Section 4.1. We appreciate the reviewer’s comment and will revise the writing in Section 4.1 to clarify the connections between these subsections at the outset.
>
> **Further clarification for Equation (4):**
>
> Equation (4) is used for computing confidence intervals used for constructing interval MDPs, and it is derived at earlier work (see Lemma 4.1). It provides a confidence bound for transitions for every state-action pair $(s,a)$, and is used to construct the interval MDPs.
> In the final version, we will include Equation (4) in the body of Lemma 4.1, and also give a brief explanation regarding the terms within Equation (4).
>
>
> **Experimental results:**
>
> As explained in Section 5, model checking for general LTL specifications over finite MDPs reduces to a reach-avoid problem over the corresponding product MDP. Therefore, regardless of the specific LTL specification, the final step always involves solving a reach-avoid problem in the product space. In response to the reviewer’s concern, we conducted an additional experiment using the same gridworld example described in the Appendix, but with a new LTL specification: “visit $G$ infinitely often while avoiding the walls.” The performance of our method remained consistent with the results reported in Figures 3, 4, and 5. Accordingly, we plan to revise the manuscript to replace the current reach-avoid task with this LTL specification. We will also include the corresponding Deterministic Rabin Automaton (DRA) and the structure of the resulting product MDP in the appendix. Finally, we would like to emphasize that the primary contribution of this paper is the development of the first regret-free controller synthesis algorithm. The experiments are primarily intended to provide insight into various aspects of the approach, including empirical performance, the gap between practical and theoretical sample complexity, and the effect of different episode lengths.
>
> Proposed changes based on comments of reviewer 3:
> - Improve writing of Section 4.1
> - Include a remark on challenges related to estimating $p_{min}$ in practice
> - Replace the current reach-avoid task with a more complex LTL specification and report the results, DRA and product MDP

---

### Official Review · Reviewer_b3Pt · 2025-03-14

**Overall Recommendation:** 3

**Summary:**

This paper tackles the problem of reinforcement learning (RL) for satisfying linear temporal logic (LTL) specifications in unknown environments modeled as Markov Decision Processes (MDPs). The authors propose what they claim is the first regret-free online RL algorithm for LTL objectives​. The approach centers on a specialized RL algorithm for infinite-horizon reach-avoid tasks (LTL “until” formulas), which is then extended to general LTL specifications by reducing them to reach-avoid problems using automata-based techniques. A separate sub-algorithm is provided to learn the underlying state-transition graph (needed for the LTL product automaton) under the assumption of a known minimum transition probability. The main contributions include rigorous finite-episode performance guarantees – in particular, a proof that the algorithm’s regret (difference in satisfaction probability compared to an optimal policy) grows only sublinearly with episodes, achieving $O(\sqrt{K})$ regret over $K$ episodes​. This translates to sharp high-confidence bounds on how close the learned controller is to optimal after a finite number of learning episodes​. In contrast to prior methods that only guarantee eventual convergence, this work provides insight into transient performance during learning. Experimentally, the paper demonstrates the algorithm on a gridworld scenario with an LTL reach-avoid goal, showing that it quickly learns an optimal policy (regret per episode drops near zero) and significantly outperforms a recent PAC-learning baseline in terms of learning speed​. Overall, the paper’s algorithmic contributions lie in combining model-based RL (optimistic exploration, episodic resets) with formal methods (automata translation of LTL) to yield the first online LTL controller synthesis method with provable regret guarantees.

**Claims And Evidence:**

The key claims of the paper are generally well-supported by the content. The claim of being the “first regret-free online algorithm” for LTL specifications appears justified. All major claims (novelty of the approach, sublinear regret, reduction to reach-avoid, need for known minimum transition probability, improved transient performance over previous methods) are either mathematically proven or empirically demonstrated. We did not find over-claiming: for instance, the authors openly acknowledge assumptions like known minimum transition probability, which has been used in previous work for identifying MECs/AMECs.

**Essential References Not Discussed:**

The paper’s reference list and related work discussion appear to cover the important literature in this area.

**Experimental Designs Or Analyses:**

The experimental evaluation, presented in the appendix (Appendix C)​, is sound and provides evidence that the algorithm works as intended. The authors test their approach on a gridworld environment with a reach-avoid LTL specification (reach a goal region G while avoiding a set of bad states B). The gridworld setup is appropriate – it’s a classic scenario to evaluate goal-reaching under uncertainty – and they introduce stochasticity by making movements succeed with 90% probability and fail (stay in place) with 10% probability​. This ensures the problem is non-trivial (the agent must deal with uncertainty). They also treat collisions with walls as entering an absorbing bad state B​, which effectively simulates an environment where hitting a wall ends the episode (a realistic “failure” condition). The experimental metrics are well-chosen to match the paper’s objectives.

**Methods And Evaluation Criteria:**

The methodology is well-chosen for the problem setting. The authors build on established RL techniques for unknown MDPs (optimistic model-based exploration in an episodic framework) and tailor them to the LTL context. In each learning episode, their algorithm constructs an interval MDP using collected data and computes an optimistic policy for a reach-avoid objective (using dynamic programming over that model)​. Importantly, they introduce an episode-specific deadline $H_k$ that serves as a maximum horizon for that episode’s execution​. This deadline mechanism, combined with a reset whenever the agent encounters a trapping state that would prevent reaching the goal, is a sensible methodological choice to handle non-communicating MDPs and ensure the agent can continue learning without getting “stuck”​. How practical this "reset button" in reality could be discussed. The evaluation criteria used in the paper align well with the methodology and objectives. The authors emphasize regret as the primary performance metric, which directly measures how well the algorithm is doing over time relative to an optimal policy. This is appropriate since the whole point is to guarantee low regret (i.e., near-optimal behavior even during learning)​.

**Other Comments Or Suggestions:**

Appendix C, “our algorithcm”

"episode length vary” should be “episode lengths vary”

Another suggestion is to provide a bit more clarification on the use of the term “sharp bounds” in the abstract​ – perhaps in the introduction or conclusion, explicitly state that the regret bound is on the order of $\sqrt{K}$ (with logarithmic factors) and that this is comparable to known lower bounds in simpler settings.

**Other Strengths And Weaknesses:**

Strengths: This work is original and significant. It tackles a known challenging gap – learning controllers for complex temporal logic goals with measurable performance during learning – and provides a novel solution with strong guarantees. Another strength is the clarity of presentation: the paper is well-structured (with a clear breakdown of the problem into subproblems), and important assumptions and definitions are stated upfront.

Weaknesses: One weakness is the strong assumption of knowing a positive lower bound on transition probabilities ($p_{\min}$). While the authors make it clear this is needed for full generality (since without it, one cannot even PAC-learn LTL​), it does raise the question of how one might obtain such a bound in practice. In very large or continuous systems, $p_{\min}$ could be extremely small or unknown. Another weakness is related to experimental validation scope. The paper demonstrates the approach on a relatively simple task (gridworld with an “until” specification). It would have strengthened the paper to see an example of a more complex LTL formula (for instance, one with an eventuality that requires the system to visit a region repeatedly, or a combination of reach-avoid tasks) to ensure the method scales to those cases. The current experiment essentially tests the core reach-avoid algorithm (Alg.1) but does not stress-test the full general LTL pipeline (Alg.4 + Alg.5) in a complex scenario.

**Questions For Authors:**

1. How does the proposed algorithm scale with respect to the size of the state space and the complexity of the LTL formula?

2. The approach requires $p_{\min}$ to be known. How realistic is this in typical applications, and what are the consequences if $p_{\min}$ is unknown? Can the authors elaborate on how one might choose or estimate $p_{\min}$ in practice?

3. In the experimental evaluation, did the authors implement the graph-learning algorithm (Alg. 5), or did they assume the transition graph was known in advance for the gridworld?

4. The experiments show a faster convergence compared to the ω-PAC method. Could the authors provide more detail on this comparison?

5. The algorithm sets an initial episode deadline $H_1$ and then adjusts it. How is $H_1$ chosen, and how robust is the algorithm to this choice?

**Relation To Broader Scientific Literature:**

The paper’s contributions are novel in the context of prior work on RL for temporal logic objectives. No previous work has achieved a regret bound for general LTL specifications in unknown MDPs – this is explicitly noted by the authors and supported by the literature​. Earlier approaches to LTL control learning fell into two broad categories: (a) Model-free or heuristic RL methods that ensured eventual convergence (with high discount factors or reward shaping) but offered no finite-sample guarantees, and (b) PAC-learning approaches that provided probabilistic guarantees on the final policy but not on the online performance. This work constitutes a natural extension of the (b) approaches by including a regret analysis.

**Theoretical Claims:**

The paper provides a series of theorems and lemmas to support its theoretical claims, and these appear to be both clear and plausible. The main theoretical result is that the proposed algorithm achieves sublinear regret. The authors give a proof sketch for this (Theorem 4.4) which outlines the key idea: classifying episodes as “fast” or “slow” and showing that slow (lengthy) episodes are rare, while fast episodes incur essentially no regret​. This reasoning is intuitive and aligns with known techniques in regret analysis for episodic MDPs – by ensuring that long exploratory episodes (which might cause more suboptimal steps) don’t happen too often, the total regret can be bounded. The proof sketch references standard concentration bounds and optimistic planning arguments, suggesting the authors are leveraging established frameworks like UCRL2 but adapted to the reach-avoid setting.

The reduction from general LTL to reach-avoid via automata is supported by known results in formal methods – the paper essentially assumes one can obtain a deterministic Rabin automaton for the LTL formula and then construct a product MDP. The graph learning algorithm (Alg. 5) is given to handle unknown transitions; its correctness is stated in terms of sample complexity (Theorem A.2 in the appendix) guaranteeing that with enough samples, the learned graph matches the true MDP’s graph with high probability​. This is a sensible approach: by knowing a lower bound $p_{\min}$, one can detect missing transitions by repeated exploration. The theoretical claim that full LTL is not PAC-learnable without such an assumption has been shown in prior work, justifying its existence.

---

> ### Author Rebuttal · Authors · 2025-03-31
>
> Thank you for appreciating the depth and technical strength of our results. We plan to improve our paper taking your comments into account. Our response to your specific questions are summarized as follows.
>
> **Computational complexity:**
>
> The primary computational steps in Algorithm 1 involve two key tasks: (1) executing the EVI algorithm, and (2) calculating $H_k$. The computational complexity of these tasks, respectively, scales quadratically and cubically, with the size of the state space. As a result, the overall computational complexity of Algorithm 1 is cubic with respect to $|S^\times| = |S| \times |Q|$, where $S$ is the state space of the MDP, and $Q$ is the state space of the DRA that corresponds to the given LTL specification.
>
> **Discussion on the practicality of known lower bounds for transition probabilities:**
>
> We agree with the reviewer that in a practice deriving $p_{min}$ might be challenging. However, all we need is to know a positive lower bound for the minimum transition probability, and not its exact value. Further, the assumption of knowing $p_{min}$ is strictly weaker than the other typical assumptions such as knowing the underlying connection graph of MDP, and there are already negative results in the literature regarding the consequences of not knowing $p_{min}$ for learning against infinite-horizon specifications. Consider a simple MDP with a state set $S = \set{s_{\text{init}}, G}$ and a singleton action set $A = \set{a}$, where the only outgoing transition from the initial state $s_{\text{init}}$ has transition probability $T(s_{\text{init}}, a, G) = \varepsilon$. It is easy to observe that, while the optimal value is one, selecting any $p_{min} > \varepsilon$ results in learning an MDP where the optimal value becomes zero. Hence, knowledge of $p_{min}$ is crucial for effectively learning in infinite-horizon tasks. In practice, domain-specific knowledge of the target system can be helpful in order to choose a sensible value for $p_{min}$.
>
>
> **Experimental results:**
> - Clarification regarding the implementation: We implemented only Algorithm 1 to obtain the results presented in the Appendix, assuming that the underlying graph structure is known. This assumption is justified by the fact that, for a fixed system dynamics, Algorithm 5 needs to be executed only once to learn the corresponding graph with the desired confidence. The resulting graph can then be reused for verifying any LTL specification. We will ensure that this assumption and its justification are stated explicitly in the final version of the paper.
>
> - Faster convergence compared to the $\omega$-PAC method: We believe that this is because our algorithm uses the intermediate confidence bounds, while the $\omega$-PAC algorithm waits until enough samples are collected, and only then starts updating its policy.
>
> - Scalability against more complex LTL specifications: As explained in Section 5, model checking for general LTL specifications over finite MDPs reduces to a reach-avoid problem over the corresponding product MDP. Therefore, regardless of the specific LTL specification, the final step always involves solving a reach-avoid problem in the product space. In response to the reviewer’s concern, we conducted an additional experiment using the same gridworld example described in the Appendix, but with a new LTL specification: "visit $G$ infinitely often while avoiding the walls". The performance of our method remained consistent with the results reported in Figures 3, 4, and 5. Accordingly, we plan to revise the manuscript to replace the current reach-avoid task with this LTL specification. We will also include the corresponding Deterministic Rabin Automaton (DRA) and the structure of the resulting product MDP in the appendix. Finally, we would like to emphasize that the primary contribution of this paper is the development of the first regret-free controller synthesis algorithm. The experiments are primarily intended to provide insight into various aspects of the approach, including empirical performance, the gap between practical and theoretical sample complexity, and the effect of different episode lengths.
>
> **Initialization for $H_1$:**
>
> Actually, we compute $H_1$, rather than starting with a choice. This is because we first run EVI to get an optimistic policy $\tilde \pi_1$ and an optimistic MDP $\mathcal{\tilde M}_1$. A DTMC with transition probability matrix $\tilde{P}_1$ is induced by fixing $\tilde \pi_1$ over $\mathcal{\tilde M}_1$, and then we use Equation (10) to derive $\tilde{Q}_1$. Finally, we use $\tilde{Q}_1$ to compute $H_1$ by using Equation (9).
>
>
> **Proposed changes based on comments of reviewer 2:**
> - Include a remark on challenges related to estimating $p_{min}$ in practice
> - Explicitly mention that the reported experimental results correspond to run of Algorithm 1
> - Replace the current reach-avoid task with a more complex LTL specification and report the results, DRA and product MDP

---

### Official Review · Reviewer_ZtQ1 · 2025-03-24

**Overall Recommendation:** 2

**Summary:**

This paper proposes a regret-free online RL algorithm for learning policies that satisfy infinite-horizon LTL specifications in unknown MDPs. The core contribution is an algorithm that, for reach-avoid specifications (a subclass of LTL), builds a sequence of optimistic policies using *interval* MDPs and extended value iteration, ensuring that the average regret—defined as the difference in satisfaction probabilities between the learned and optimal policies—converges to zero. The authors then extend this to general LTL specifications by transforming the original problem into a reach-avoid problem over a product MDP composed with a deterministic Rabin automaton, using an auxiliary algorithm to learn the graph structure of the unknown MDP given a known lower bound on transition probabilities. They provide theoretical regret bounds and claim sublinear regret in the number of episodes, supported by a regret decomposition analysis. Experimental results in the appendix in a gridworld domain are presented to suggest improved sample efficiency over a prior PAC-MDP approach.

**Claims And Evidence:**

While the paper presents a complete pipeline and claims theoretical guarantees, the mathematical formulation and algorithmic descriptions are occasionally imprecise, and the overall presentation order may obscure key assumptions or derivations. Hence, the writing and proofs provided in support of the claims are not clear and convincing.

**Essential References Not Discussed:**

The paper extensively covers related works.

**Experimental Designs Or Analyses:**

N/A

**Methods And Evaluation Criteria:**

Yes.

**Other Comments Or Suggestions:**

- No explanation is given for the right hand side of Equation 4 (not even an intuitive/brief one). E.g Why $8|S| log(2|A|k/3δ)$ in the numerator? Why $max (1,N_k(s, a))$ in the denominator? Why the square-root?
- It is unclear what is an optimistic MDP $\mathcal{\tilde M}\_k \in \mathbf{\mathcal{M}}\_k$ . Please define it.
- Remark 4.3. says "One may notice that the set B also includes every MEC whose intersection with G is empty.". How can this be true given that B is a set of states and a MEC is an MDP (from the definition in Sec 3)?
- Cite the definition of MDPs used in the preliminaries section, or explicitly mention how it is different from a typical definition (cite).
- cite maximal end components (MEC)
- Cite extended value iteration (EVI)
- Define and cite what is an Interval Markov Decision Processes
- Line 410 should use $T^\times : S^\times \times A^\times \times S^\times$ instead of  $T^\times : S^\times$

**Other Strengths And Weaknesses:**

The paper provides a comprehensive number of algorithms and theoretical results. However, it is extremely unclear in its writing making it hard to follow and evaluate its correctness.

- The algorithms are not clearly explained. For example, at the start of Section 4.1 the authors mention "Alg. 1 shows our learning algorithm.". But Alg. 1 cannot be understood at that point since it references Alg. 2, Eq. 4, and Eq. 9 which only appear later (where Alg. 2 and Eq. 4 are also not clearly explained). Additionally, it is unclear what the notation #$\\{t < t_k : s_t = s, a_t = a\\}$ means (same for  #$\\{t < t_k : s_t = s, a_t = a, s_{t+1}=s'\\}$). Is it a set of only one Boolean value (the result of $t < t_k$)?

-  Alg. 2 and  Alg. 3 are not clearly explained. In fact,  Alg. 2 uses  Alg. 3, but  Alg. 3 is not even referenced in the text. It would have helped if the authors clearly explained their extended value iteration, and how exactly they obtain a policy that maximizes the probability of reaching G.

Finally, this makes even the main algorithm (Alg. 4) unclear since it uses Alg. 1, and makes the correctness of the corresponding main theorem (Theorem 5.1) hard to evaluate. Hence, while I think the problem this paper attempts to solve is very relevant, and the paper offers numerous interesting algorithms and theoretical results, its presentation is currently too unclear.

# Post Rebuttal

The authors rebuttal did help a bit clarify the assumptions they make in this paper, their definition of $\mathbb{P}(.)$ (how it differs from $Pr$), where Lemma 4.1 comes from, and the explanation for Equation (4). Unfortunately, I still have reservations regarding the clarity of the paper (listed below). However, I can see that the other reviewers feel positive about the paper, so it is possible that I missed something.

- It is still unclear how the authors can simply use Lemma 3 from Tarbouriech et al. (2020) if the authors are not assuming SSP-communicating MDPs. Perhaps I am misunderstanding the assumptions of that Lemma. Additionally, it is unclear how that Lemma is the same as Lemma 4.1 when:

    - $\mathcal{E}$ from Lemma 4.1 is defined differently from $\mathcal{E}$ in Tarbouriech et al. Lemma 3. If one follows from the other, no explanation is given on how.
    - For a given $\delta \in(0,1)$, the bound from Tarbouriech et al. Lemma 3 is $\mathbb{P}(\mathcal{E})\geq1-\frac{\delta}{3}$ but the one in Lemma 4.1 is $\mathbb{P}(\mathcal{E})\geq 1-\delta$.

- If the authors are not assuming SSP-communicating MDPs or the existence of a proper policy, then it seems like $\Lambda(s)$ and $\lambda^*(s)$ can be unbounded for some $s$, invalidating Lemma 4.2. For example, Consider a simple MDP with a state set $S = \set{s_{\text{init}}, s_1,s_2, G}$ and a singleton action set $A=\\{a\\}$. The transition probabilities are $T(s_{\text{init}}, a, G) = p_{min}$, $T(s_{\text{init}}, a, s_1) = 1-p_{min}$, $T(s_1, a, s_2) = 1-p_{min}$, and $T(s_2, a, s_1) = p_{min}$. Then $\Lambda(s)$ and $\lambda^*(s)$ are unbounded.

- The authors still did not clarify the notation #{.} in their rebuttal (they did not even confirm or denied the interpretation I gave), and in general they did not clarify my concerns with Algorithm 1. As I already highlighted in my review, I didn't find the presentation of Algorithm 1 nor the explanations given Section 4.1 entirely clear. Hence it did not help when they simply referred me back to Section 4.1 without an attempt to clarify at least some of those concerns (e.g. regarding how their EVI is able to find a policy that maximizes the probability of reaching G).

In general, I think the paper makes sense at a high level, and the specific algorithms are very interesting and do look like they work. However, given the focus of this paper theoretical guarantees, the impreciseness and potentially incomplete assumptions are problematic. Hence, I am maintaining my score for the moment but I am happy to update it if the other reviewers feel differently about these outstanding concerns.

**Questions For Authors:**

Please refer to my weaknesses and comments above.

**Relation To Broader Scientific Literature:**

The paper builds on prior work in reinforcement learning with temporal logic specifications, particularly extending ideas from PAC-MDP and UCRL2 frameworks by aiming for regret bounds rather than asymptotic guarantees. It distinguishes itself from earlier approaches by handling non-communicating MDPs and providing better finite-time performance guarantees (regret of $O(K^{1/2})$) than the closest related prior work (regret of $O(K^{3/2})$).

**Theoretical Claims:**

- Lemma 4.1. (Tarbouriech et al., 2020) is unclear since:
  - It is unclear what is the specific Lemma/Theorem from Tarbouriech et al. (2020) that is being restated here.
  -  $\mathbb{P}(\mathcal{E})$ is undefined (note that the authors generally use the notation $Pr$ to refer to probability distributions).

-  The proofs of most of the Lemmas and Theorems use the results of Tarbouriech et al. (2020), but that means the authors must assume that the MDPs are SSP-communicating. However, this assumption is not stated in the paper (neither in *Problem 1* nor *Problem 2* statements). It is unclear what other specific assumptions are made but not stated.

---

> ### Author Rebuttal · Authors · 2025-03-31
>
> We thank you for the detailed and useful feedback! We plan to improve our paper taking your comments into account. Our response to your specific questions are summarized as follows.
>
> **Assumptions used in the paper:**
> We do not assume the SSP-communicating property for MDPs. Instead, we **only** make a weaker assumption of knowing a non-zero lower bound over minimum transition probability $p_{min}$, which enables application of our method to the general non-communicating MDPs​. To relax the communicating assumption in Tarbouriech et al. (2020), we leverage $p_{min}$​ to compute $\Lambda(s)$, which represents an upper bound on the minimum expected time required to reach $G$ from state $s$ in the (artificial) MDP $\mathcal{M}'$, constructed by connecting $B$ to $𝑠_{𝑖𝑛𝑖𝑡}​$. Note that $p_{min}$ can be used to compute the underlying graph to any desired accuracy, enabling us to check whether $G$ is reachable (which implies the boundedness of $\Lambda(s)$). These details are discussed in Lemma 4.2 and Remark 4.3. In particular, our results require **no** assumption other than knowing $p_{min}$. Also, it should be noted that assuming knowledge of $p_{min}$ is strictly weaker than the assumption of SSP-communication.
>
>
> **Presentation of Algorithm 1:**
> Regarding referencing other algorithms and equations, we would like to highlight that Section 4.1 is specifically dedicated to explaining various components of the main algorithm (Algorithm 1). In particular, the following components of Algorithm 1 are discussed in detail: (1) the application of Equation (4) for “computing confidence intervals”, which are then used to construct interval MDPs from the observed data; (2) the operation of our proposed “Extended Value Iteration” method (Algorithm 2); and (3) the use of Equation (9) to compute “episode-specific deadlines”. These explanations are provided in the respective subsections of Section 4.1. That said, we appreciate the reviewer’s comment and will, in the final version of the paper, provide additional intuitive explanations of our Extended Value Iteration (Algorithm 2), include a brief description of Algorithm 3, and define the notation \#{.}, which denotes set cardinality.
>
>
> **Clarity of Lemma 4.1:**
> In Lemma 4.1, the event $\mathcal{E}$ refers to the scenario where the actual MDP lies within the interval MDP computed during the $k^{th}$ episode of learning. The notation $\mathbb{P}(\mathcal{E})$ represents the probability that this event occurs. The notation $Pr_{s_{init}}^{\pi}[\varphi]$ is explicitly defined in Section 3 under the heading “Maximum Probability of Satisfaction”, and refers to the probability of satisfying a given LTL specification $\varphi$, starting from the initial state $s_{init}$ and following policy $\pi$. In the camera-ready version of our paper, we will ensure that the notation $\mathbb{P}(.)$ is clearly explained, replace instances of $Pr[.]$ by $\mathbb{P}(.)$, and we will explicitly reference the relevant lemma adapted from (Tarbouriech et al., 2020, Lemma 3).
>
>
> **Comment on missing explanation for Equation (4):**
> Equation (4) is used for computing confidence intervals necessary for constructing interval MDPs, and it is derived in earlier work (see Lemma 1). Specifically, the derivation relies on well-known probabilistic inequalities, and $ max(1, N_k(s,a))$ serves to prevent division by zero, where $N_k(s,a)$ represents the number of visits to the state-action pair $(s, a)$. In the final version, we will include Equation (4) within the body of Lemma 4.1 and provide a brief explanation of the terms in Equation (4).
>
>  **Proposed changes based on comments of reviewer 1:**
> - Provide additional intuitive explanation of our EVI (Algorithm 2), and give a brief explanation of Algorithm 3
> - Integrate Equation (4) into Lemma 4.1, and provide a brief explanation of the terms within Equation (4)
> - Explain the notation $\mathbb{P}(.)$ under the notations section
> - Define the notation \#{.}
> - Provide citations for the definition of MDPs and interval MDPs (IMDPs), maximal end components (MECs), extended value iteration (EVI), and refer to the specific lemma from (Tarbouriech et al., 2020) for Lemma 4.1
> - Define optimistic MDPs

---

### Decision · Program_Chairs · 2025-05-01

**Decision:**

Accept (poster)

**Comment:**

This paper proposes a regret-free reinforcement learning algorithm for satisfying LTL specifications in unknown MDPs. The approach begins with a method for infinite-horizon reach-avoid tasks and extends to general LTL objectives via automata-based reductions. A core contribution is the theoretical guarantee of sublinear regret in the number of episodes, supported by formal proofs and empirical evaluation in a gridworld domain.

All reviewers agree that the paper makes a significant theoretical contribution by being the first to provide regret bounds for LTL-constrained RL. The proof of sublinear regret is well-grounded in standard RL techniques, under the assumption of a known lower bound on transition probabilities ($p_{min}$). While the experiments are limited to a basic tabular gridworld, they demonstrate the algorithm’s superior sample efficiency compared to a PAC-learning baseline. The assumption of $p_{min}$ is noted as a potential limitation in practical scenarios. Reviewers also expressed concerns about the clarity of Section 4.1 and Algorithm 1. The authors are encouraged to clarify how their analysis builds upon Tarbouriech et al. (2020), particularly in the key proofs, and to clearly explain how the assumption of SSP-communicating MDPs is relaxed in their work.

Overall, the paper makes a compelling and novel contribution to LTL-constrained RL by introducing a regret-based framework with rigorous theoretical support. Despite some concerns about presentation clarity, the work substantially advances the field. The meta-reviewer encourages the authors to revise the manuscript based on the reviewers' feedback.